# Towards Customized Knowledge Distillation for Efficient Dense Image Predictions

**Dong Zhang**                                                    *dongz@ust.hk*
*Department of Electronic and Computer Engineering*
*The Hong Kong University of Science and Technology*

**Pingcheng Dong**                          *pingcheng.dong@connect.ust.hk*
*Department of Electronic and Computer Engineering*
*The Hong Kong University of Science and Technology*

**Long Chen**                                               *longchen@cse.ust.hk*
*Department of Computer Science and Engineeringk*
*The Hong Kong University of Science and Technology*

**Kwang-Ting Cheng**                                        *timcheng@ust.hk*
*Department of Electronic and Computer Engineering*
*The Hong Kong University of Science and Technology*

**Reviewed on OpenReview:** *https://openreview.net/forum?id=4verIe3tE4*

## Abstract

It has been revealed that efficient dense image prediction (EDIP) models designed for AI chips, trained using the knowledge distillation (KD) framework, encounter two key challenges, including maintaining boundary region completeness and ensuring target region connectivity, despite their favorable real-time capacity to recognize the main object regions. In this work, we propose a customized boundary and context knowledge distillation (BCKD) method for EDIPs, which facilitates the targeted KD from large accurate teacher models to compact small student models. Specifically, the boundary distillation focuses on extracting explicit object-level boundaries from the hierarchical feature maps to enhance the student model's mask quality in boundary regions. Meanwhile, the context distillation leverages self-relations as a bridge to transfer implicit pixel-level contexts from the teacher model to the student model, ensuring strong connectivity in target regions. Our method is specifically designed for the EDIP tasks and is characterized by its simplicity and efficiency. Extensive experimental results across semantic segmentation, object detection, and instance segmentation on five representative datasets demonstrate the effectiveness of BCKD, resulting in well-defined object boundaries and smooth connecting regions.

## 1 Introduction

The dense image prediction (DIP) tasks, *e.g.*, semantic segmentation (Long et al., 2015), object detection (Girshick, 2015), and instance segmentation (Wang et al., 2021c), are challenging problems within both domains of computer vision and multimedia computing (Zhang et al., 2020; Ahn et al., 2019). The objective of these tasks is to assign a semantic label to each object and/or pixel of the given image (Zhang et al., 2020). In recent years, achievements in general-purpose GPU technology have resulted in notable enhancements in both size and accuracy of sophisticated DIP models (Cao et al., 2022a; Strudel et al., 2021), *e.g.*, Mask2Former (Cheng et al., 2022), SegNeXt (Guo et al., 2022b), and SAM (Kirillov et al., 2023). However, deploying these large and accurate DIP models on resource-constrained edge computing devices, *e.g.*, artificial intelligence chips (Dong et al., 2025), presents significant challenges due to the substantial computational costs and high memory consumptions associated with these models (Wang et al., 2021c).

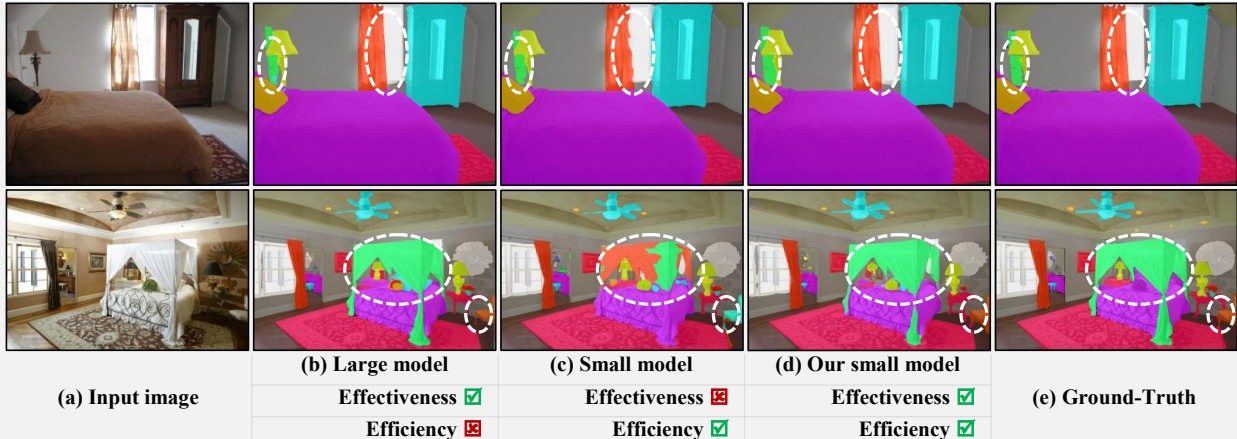

Figure 1: Two representative cases that small models are prone to predict errors. Visualization comparisons between large accurate models (b) and small efficient models (c) show that the latter tend to make errors in maintaining boundary region completeness and preserving target region connectivity. With the help of BCKD, our small models in (d) can address the two types of errors, leading to crisp region boundaries and smooth connecting regions. Samples are from the ADE20K dataset (Zhou et al., 2017).

Compressing large DIP models into compact efficient dense image prediction (EDIP) models offers an intuitive and cost-effective solution to address the severe resource limitations associated with mapping vision models onto edge computing devices (Dong et al., 2021; Zhang et al., 2021). In particular, the cross-architecture fashion enables compressed models to seamlessly adapt to customized edge chips, eliminating the need for hardware modifications while maintaining computational efficiency. This manner significantly reduces deployment complexity and enhances the flexibility of model inference across heterogeneous edge computing platforms (Yin et al., 2023). To achieve this goal and develop accuracy-preserving EDIP models, knowledge distillation (KD) (Hinton et al., 2015; Wang et al., 2023), a prevalent model compression technology, has been pragmatically employed by using a small efficient model (*i.e.*, the student model) by imitating the behavior of a large accurate model (*i.e.*, the teacher model) in training (Hinton et al., 2015; Zhao et al., 2022). During inference, only the student model is utilized, allowing for a highly-efficient recognition pattern while simultaneously reducing the model size (Dong et al., 2021; Zhang et al., 2021; Cui et al., 2023). Despite significant advancements by current KD methods across multiple dimensions, including sophisticated distillation strategies (Cui et al., 2023) and complex distillation content (Wang et al., 2020), the inherent complexity of DIP continues to pose two critical challenges for existing approaches, particularly for the efficient compact models.

Primary KD methods mainly emphasize the imitation of general knowledge (*e.g.*, features, regions, and logits) while overlooking the nuanced understanding of features along objective semantic boundaries and connecting internal regions essential for EDIPs (Zhang et al., 2021; Wang et al., 2022a). Particularly, since the small student model often predicts the main object regions fairly well but fails in boundary and connecting regions (Fu et al., 2019; Yuan et al., 2020; Cao et al., 2022a), the conventional utilization of task-agnostic general KD may not be effective enough and can be considered purposeless and redundant (Gou et al., 2021; Xu et al., 2020; Zhao et al., 2022), remaining a performance gap between the obtained results and the expected ones (Wang et al., 2024). For instance, we recommend the representative semantic segmentation task as an example. As shown in Figure 1, the small student PSPNet-18 model in (c) produces inferior results compared to the large teacher PSPNet-101 model results in (b). The student model wrongly segments the boundary regions of *"curtain"* and *"door"* as the background category or other foreground objects, and produces fragmented *"bed valance"* and *"chair"*, breaking the regional relation connectivity. Generally, the common errors observed in the outputs of the small student model can be summarized as maintaining boundary region completeness and ensuring target region connectivity.

To mitigate these errors and narrow the performance gap, we propose a customized and targeted KD strategy termed as **B**oundary and **C**ontext **K**nowledge **D**istillation (BCKD). By "customized", we mean that our method's inherent ability to synergistically address the common errors present in existing EDIP models,

while is also generally compatible with other methods (*ref.* Sec. 5.3). BCKD mainly consists of two key components: the boundary distillation and the context distillation, aimed at rectifying the typical common errors encountered by EDIP models in maintaining boundary region completeness and ensuring target region connectivity, respectively. Specifically, boundary distillation involves generating explicit object-level boundaries from the hierarchical backbone features, enhancing the completeness of the student model's masks in boundary regions (*ref.* Sec. 4.2). At the same time, context distillation transfers implicit pixel-level contexts from the teacher model to the student model through self-relations, ensuring robust connectivity in the student's masks (*ref.* Sec. 4.3). BCKD is tailored specifically for EDIP tasks and offers a more targeted distillation pattern and a more tailored distillation manner compared to conventional task-agnostic KD methods. From a rigorous theoretical perspective, we establish and prove the effectiveness of our BCKD method (*ref.* Sec. A). To validate the accuracy, we conducted extensive experiments in three representative dense image prediction tasks, including semantic segmentation, object detection, and instance segmentation, utilizing five challenging datasets such as Pascal VOC 2012 (Everingham et al., 2010), Cityscapes (Cordts et al., 2016), ADE20K (Zhou et al., 2017), COCO-Stuff 10K (Caesar et al., 2018), and MS-COCO 2017 (Lin et al., 2014). Qualitatively, BCKD produces sharp region boundaries and smooth connecting regions, addressing challenges that hindered existing EDIP models. Quantitatively, BCKD consistently improves the accuracy of baseline models in various metrics, achieving competitive performance.

The main contributions of this work are: *(1)* We revealed two prevalent issues in existing EDIP models: maintaining boundary region completeness and ensuring target region connectivity. *(2)* We proposed a customized and targeted boundary and context knowledge distillation method, which not only demonstrates inherent coherence, but is also generally compatible with other methods. *(3)* Experimental evaluations across various tasks, baselines and datasets illustrate a new state-of-the-art accuracy of our method in comparison with existing methods.

## 2 Related Work

### 2.1 Dense Image Prediction Tasks

Dense image prediction (DIP) is a fundamental research problem within the fields of computer vision and multimedia computing, with the objective of assigning each object and/or pixel in an input image to a predefined category label, thereby enabling comprehensive semantic image recognition (Long et al., 2015; Zhou et al., 2024; Zhang et al., 2020; Lin et al., 2023b). Current mainstream DIP models can be roughly classified into the following three categories based on their backbone components: 1) methods based on CNNs (Long et al., 2015; Yu et al., 2018; Noh et al., 2015; Huang et al., 2019), 2) methods based on ViT[1] (Strudel et al., 2021; Wang et al., 2022b; Zheng et al., 2021), and 3) methods that combine CNNs and ViT (Li et al., 2022a; Peng et al., 2021). The key difference between these types of architectures is the approach used for feature extraction and how the extracted features are utilized in enhancing the capacity of CNNs models to capture contextual features (Cao et al., 2022a; Zhang & Cheng, 2025), increasing the capacity of ViT models to capture local features (Wu et al., 2021; Zhang et al., 2022b; Peng et al., 2021), and leveraging low-level features to improve representation capacity (Zhang et al., 2023; Xie et al., 2021) for achieving favorable results. Concretely, due to differences in feature extraction manners between CNNs and ViT, these two categories exhibit slight performance differences (Wang et al., 2022b; Peng et al., 2021; Zheng et al., 2021; Li et al., 2022a). For example, CNNs methods are better at predicting local object regions, while ViT methods, due to their stronger contextual information, can produce more complete object masks. Fortunately, the mixture of CNNs and ViT (*e.g.*, CMT (Guo et al., 2022a), CvT (Wu et al., 2021), ConFormer (Peng et al., 2021), CAE-GreaT (Zhang et al., 2023), and visual Mamba (Gu & Dao, 2023)) uses the representation strengths of both patterns, resulting in highly satisfactory recognition performance (Han et al., 2022; Mao et al., 2022; Wang et al., 2021a). In addition to these fundamental categories, there are advanced approaches that also utilize task-specific training tricks (*e.g.*, graph reasoning (Zhang et al., 2022b), linear attention (Kong et al., 2022), mult-scale representation (Fan et al., 2021)) to improve the accuracy. However, while current

---

[1]We consider the visual state space model-based methods as a specialized Transformer architecture (Gu & Dao, 2023; Xu et al., 2024), owing to its structural similarities with the ViT model. Besides, we do not address the content related to these models. Therefore, we will no longer have separate discussions on this aspect.

methods have achieved promising accuracy, mapping these models on resource constrained edge computing devices remains challenging because these devices typically have limited computation resources and memory consumptions (Dong et al., 2023; 2025). In this work, we do not intend to modify the network architecture. We first investigate the result disparities between small and large models and then propose a novel KD strategy tailored to the EDIP models. We aim at improving the recognition accuracy of small models without requiring any extra training data or increasing the inference costs.

## 2.2 Knowledge Distillation in DIPs

Knowledge Distillation (KD) is a well-established model compression technology that aims to transfer valuable knowledge from a large accuracy teacher model to a small efficient student model, with the objective of enhancing the student's accuracy during inference (Chen et al., 2023; Gou et al., 2023; Xiang et al., 2025; Wang et al., 2020; Xu et al., 2025; Li et al., 2025). It is worth mentioning that the effectiveness of KD in cross-architecture scenarios has enabled significant flexibility in artificial intelligence chip design, as it eliminates the need to modify the underlying operators while maintaining model performance, which provides a practical solution for hardware adaptation without compromising computational efficiency (Dong et al., 2025). The key factors for the success of KD in DIPs are: 1) the types of knowledge being distilled, *e.g.*, general knowledge: features and logits, and task-specific knowledge: class edge for semantic segmentation and object localization for object detection, 2) the distillation strategies employed, *e.g.*, offline distillation (Tseng et al., 2022), online distillation (Guo et al., 2020), and self-distillation (Zhang et al., 2019), and 3) the architecture of the teacher-student pair, *e.g.*, multi-teacher KD (Yuan et al., 2021), attention-based KD (Passban et al., 2021), and graph-based KD (Lee & Song, 2019). While the effectiveness of existing KD methods has been validated in general vision tasks, current methods rely on coarse task-agnostic knowledge and do not consider the task-specific feature requirements (Sun et al., 2020; Gou et al., 2021; Mao et al., 2022; Chen et al., 2017a; Wang et al., 2019). Especially for EDIPs, models are highly sensitive to feature representations (Long et al., 2015; Zhang et al., 2022c; Zheng et al., 2021; Zhang et al., 2018). Therefore, general KD may not be effective enough and can be considered purposeless and redundant (Gou et al., 2021; Xu et al., 2020; Zhao et al., 2022; Zhang et al., 2022b), remaining a performance gap between the obtained results and the expected ones (Yang et al., 2022a; Cui et al., 2023; Wang et al., 2024). Recent studies have shown that task-specific patterns of KD can help further improve the performance of student models (Zheng et al., 2022). For example, in object detection, object localization KD leads to more accurate predictions than the general knowledge (Sun et al., 2020; Zhixing et al., 2021). In this work, we also adopt the idea of task-specific KD. Our contribution lies in proposing a customized KD scheme, namely boundary distillation and context distillation, which target the common errors of EDIP models, namely their tendency to make errors in maintaining boundary region completeness and ensuring target region connectivity. It is also worth noting that while some methods, *e.g.*, CTO (Lin et al., 2023c), SlimSeg (Xue et al., 2022), and BPKD (Liu et al., 2024), have integrated boundary information in EDIPs, they necessitate the pre-extraction and incorporation of ground-truth boundaries (Xue et al., 2022; Liu et al., 2024; Lin et al., 2023c). In contrast, our method obviates the demand for pre-extracting boundaries, thereby making it more practical for real-world applications and enabling savings in time and labor.

## 3 Preliminaries

In the training phase, KD intends to facilitate expectant knowledge transfer from a large teacher model $\mathbb{T}$ to a small compact student model $\mathbb{S}$, with the primary goal of enhancing the accuracy of $\mathbb{S}$ (Hinton et al., 2015; Zhang et al., 2021; 2019; Gou et al., 2021; Ji et al., 2021). In inference, only $\mathbb{S}$ is used, so there are no computational overheads. Typically, features and logits serve as a medium for knowledge transfer. Besides, the temperature scaling strategy is often utilized to smooth the features and logits, which helps to lower prediction confidence and alleviate the issue of excessive self-assurance in $\mathbb{T}$ (Phuong & Lampert, 2019). Formally, KD can be expressed by minimizing the cross-entropy loss as follows:

$$\mathcal{L}_{KD} = -\tau^2 \sum_{i \in M} \sigma(\mathbf{T}_i)^{1/\tau} \log\left(\sigma(\mathbf{S}_i)^{1/\tau}\right), \tag{1}$$

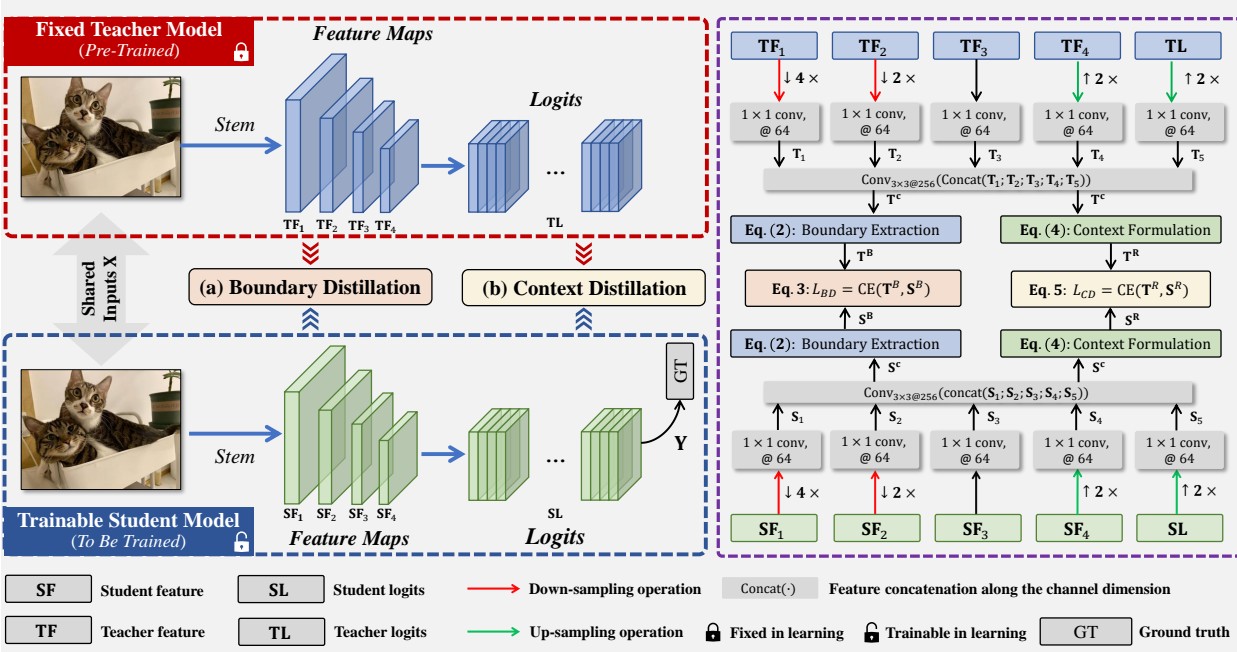

Figure 2: The overall architecture of our boundary and context distillation strategy for the efficient dense image prediction tasks, where the right side illustrates the implementation details of the whole network. Specifically, the boundary distillation involves generating explicit object-level boundaries from the hierarchical backbone features, enhancing the completeness of the student model's masks in boundary regions (*ref.* Sec. 4.2). Meanwhile, the context distillation transfers implicit pixel-level contexts from the teacher model to the student model through self-relations, ensuring robust connectivity in the student's masks (*ref.* Sec. 4.3). Compared to existing methods, our method demonstrates a stronger specificity for EDIP and inherently possesses the ability to synergistically address the common errors found in small models.

where $\mathbf{T}_i$ and $\mathbf{S}_i$ (both have been adjusted to the same dimension via $1 \times 1$ convolutions) are the $i$-th feature/logit item extracted from $\mathbb{T}$ and $\mathbb{S}$, respectively. $\sigma(\cdot)$ is the softmax normalization operation along the channel dimension, and $\tau \in \mathbb{R}_+$ denotes the temperature scaling coefficient. $M$ denotes the learning objective of $\mathbf{T}_i$ and $\mathbf{S}_i$, which typically refers to the spatial dimensions. Following (Zhang et al., 2019; Phuong & Lampert, 2019), to simplify temperature scaling effectively, we use $\mathbf{T}_i/\mathbf{S}_i$ divided by $\tau$ to achieve the similar effect. In addition to cross-entropy loss, other loss functions are also commonly used for KD, including KL divergence loss and MSE loss (Liu et al., 2019; 2022; Zheng et al., 2023). While existing KD methods have demonstrated promising results across various vision tasks (Zhang et al., 2021; Cui et al., 2023; Xu et al., 2020; Liu et al., 2019; Wang et al., 2020; Gou et al., 2021), they have not adequately addressed the specific feature understanding required for object boundaries and connecting regions in EDIP tasks. This oversight has led to suboptimal performance in these contexts. In following, we will introduce a complementary and targeted KD scheme informed by the common failure cases observed in small models, as shown in Figure 1, with the aim of enhancing their inference accuracy.

# 4 Customized Knowledge Distillation

## 4.1 Overview

Figure 2 illustrates an overview of the network architecture for our proposed BCKD. The whole network mainly consists of an accurate teacher network $\mathbb{T}$, which is a large network that has been trained, and a small efficient network $\mathbb{S}$ that is waiting to be trained. The input for $\mathbb{T}$ and $\mathbb{S}$ is an arbitrary RGB image $\mathbf{X}$, and the output of $\mathbb{S}$ is a semantic mask or bounding box $\mathbf{Y}$ that predicts each pixel or/and object with a specific class label. The hierarchical features extracted from the backbone network are concatenated along the

channel dimension to facilitate the extraction of EDIP-specific boundary and contextual information from $\mathbf{X}$. The concatenated features with 256 channel dimension are defined as $\mathbf{T}^c = \text{Conv}_{3\times3}(\text{concat}(\mathbf{T}_1; \mathbf{T}_2; \cdots; \mathbf{T}_5))$ and $\mathbf{S}^c = \text{Conv}_{3\times3}(\text{concat}(\mathbf{S}_1; \mathbf{S}_2; \cdots; \mathbf{S}_5))$ for $\mathbb{T}$ and $\mathbb{S}$, respectively. It should be noted that both $\mathbf{T}_i$ and $\mathbf{S}_i$ that are concatenated have been uniformly resized into $1/8$ of $\mathbf{X}$'s spatial size via $1 \times 1$ convolution and up-/down-sampling operations. In training, we propose targeted boundary distillation and context distillation strategies that are tailored for the EDIP tasks. Specifically, the boundary distillation synthesizes explicit object-level boundaries $\mathbf{T}^B$ and $\mathbf{S}^B$ from $\mathbf{T}^c$ and $\mathbf{S}^c$, respectively, thereby the completeness of $\mathbb{S}$'s results in the boundary regions can be enhanced. At the same time, the context distillation transfers implicit pixel-level relations $\mathbf{T}^R$ and $\mathbf{S}^R$ by using self-relations, ensuring that $\mathbb{S}$'s results have strong target region connectivity.

## 4.2 Boundary Distillation

The semantic object boundary is defined as a set of arbitrary pixel pairs from the given image, where the boundary between pairwise pixels has a value of 1 if they belong to different classes, while if the pairwise pixels belong to the same class, the boundary between them has a value of 0 (Ahn & Kwak, 2018). Moreover, this attribute also exists in the hierarchical features/logits extracted by the backbone feature maps (Chen et al., 2020). In our work, we use the semantic affinity similarity between arbitrary pairwise pixels from $\mathbf{T}^c$ or $\mathbf{S}^c$ to obtain the explicit object-level boundaries (Ahn et al., 2019; Ru et al., 2022). Concretely, for a pair of image pixels $\mathbf{T}_i^c$ and $\mathbf{T}_j^c$, $\mathbf{T}_{i,j}^B$ can be formulated as:

$$\mathbf{T}{i,j}^B = 1 - \max p, q \in \Pi_{i,j} \mathcal{B}\left(\text{Conv}_{1\times1}(\mathbf{T}^c p), \text{Conv}_{1\times1}(\mathbf{T}^c q)\right), \tag{2}$$

where $\mathbf{T}_p^c$ and $\mathbf{T}_q^c$ are two arbitrary pixel items from $\mathbf{T}^c$, and $\Pi_{i,j}$ denotes a set of pixel items on the line between $\mathbf{T}_i^c$ and $\mathbf{T}_j^c$. $\text{Conv}_{1\times1}$ denotes a $1\times1$ convolution layer that is used to compress the channel dimension, where the input channel size is 256 and output channel size is 1. $\mathcal{B}(\cdot)$ denotes the operation that determines the object-level image boundary values, which outputs either 1 or 0. $\mathbf{T}^B$ can be obtained across the entire spatial domain, and $\mathbf{S}^B$ can be obtained analogously. Based on $\mathbf{T}^B$ and $\mathbf{S}^B$, boundary distillation loss is formulated as:

$$\mathcal{L}_{BD} = -\tau^2 \sum_{i \in M} \rho(\mathbf{T}_i^B)^{1/\tau} \log\left(\rho(\mathbf{S}_i^B)^{1/\tau}\right), \tag{3}$$

where $\rho$ denotes the spatial-wise softmax normalization. By Eq. (3), $\mathbb{T}$'s accurate prediction of the object boundary region can be transferred into $\mathbb{S}$, thereby addressing its issue of maintaining boundary region completeness.

The proposed boundary distillation strategy can effectively address the limitations of EDIP models in achieving completeness in boundary region predictions. It is important to highlight that while several advanced methods, such as BGLSSeg (Zhou et al., 2024), SlimSeg (Xue et al., 2022), and BPKD (Liu et al., 2024), leverage explicit edge information for semantic segmentation, these methods necessitate the pre-extraction and integration of ground-truth masks (please refer to Table 3 for further details). More importantly, the extracted edge information often lacks the semantic context of the objects and may introduce noise (Yang et al., 2022b; Ji et al., 2022). In contrast, our method eliminates the need for pre-extracting image edges. By utilizing hierarchical feature maps, our method enhances the delineation of object boundaries with more comprehensive semantic information while mitigating the adverse effects of noise present in shallow features. As illustrated in Figure 3, we show a visualization comparison of the extracted image boundaries between our method and the ground truth edge method used in the state-of-the-art BPKD (Liu et al., 2024) model. We can observe that the semantic boundaries extracted by our method can better cover the actual boundaries of the semantic objects without introducing background noise information. This innovation not only can enhances the practicality of our method for real-world applications but also streamlines the overall process, leading to reductions in both time and labor requirements.

## 4.3 Context Distillation

Elaborate object relations, as validated in (Caron et al., 2021; Wang et al., 2021b; Li et al., 2022b), are beneficial for learning implicit contextual information across spatial dimensions (Lin et al., 2023a; Ji et al., 2022). In this paper, we also adopt this scheme to address the challenge of inadequate preservation of target

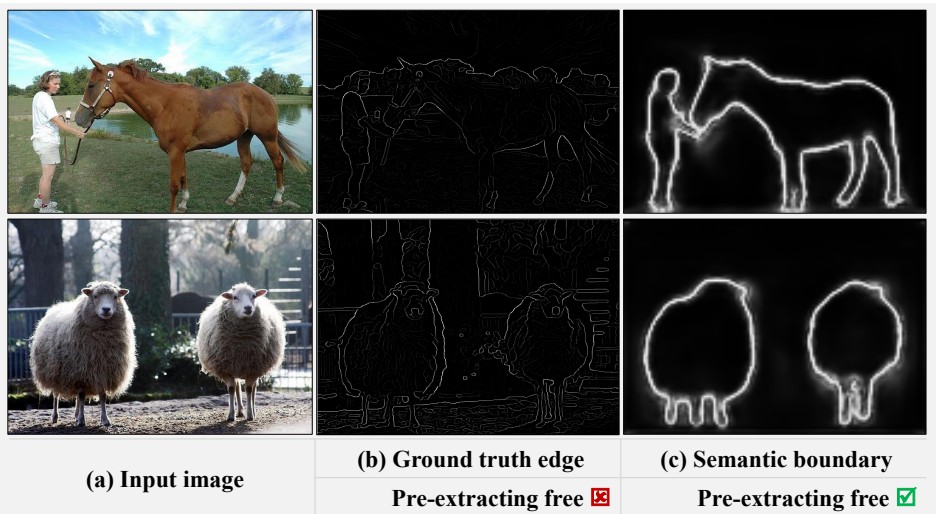

| (a) Input image | (b) Ground truth edge | (c) Semantic boundary |
| | Pre-extracting free ❌ | Pre-extracting free ☑ |

Figure 3: The visualization comparisons of the extracted image boundaries between our method in (c) and the ground truth edge method in (b) used in the state-of-the-art BPKD Liu et al. (2024). Images are from Pascal VOC 2012 Everingham et al. (2010).

region connectivity in EDIPs. We consider this scheme as the medium in the KD process. Our contribution lies in treating pixel-level relations as a bridge for context transfer. Compared to existing methods (Li et al., 2022b; Ji et al., 2022), our approach utilizes pixel-level relations solely during the training process, thereby avoiding the increase in model complexity and parameters in inference that is typically associated with current methods (Liu et al., 2019; Lin et al., 2023a). Besides, compared to object-level relations, the employed pixel-level relations can capture global contextual information more comprehensively, making them more suitable for DIP tasks. Specifically, for the concatenated features $\mathbf{T}^c$ of $\mathbb{T}$, its self-relation $\mathbf{T}^R$ is formulated as:

$$\mathbf{T}^R = \sigma \left( \frac{O(\mathbf{T}^c)^T \cdot O(\mathbf{T}^c)}{\sqrt{d}} \right) / \tau \in \mathbb{R}^{hw \times hw}, \tag{4}$$

where $O(\cdot)$ denotes the feature-aligned operation as in (Li et al., 2022b; Xie et al., 2023), which aims to align the feature distribution of the $\mathbb{S}$ with that of the $\mathbb{T}$ as closely as possible. $O(\cdot)$ is implemented as a projection head consisting of two $1 \times 1$ convolutional layers (256 channels) with ReLU activation, followed by layer normalization and final projection to the target dimension. $T$ denotes the matrix transpose operation. $d$ is the channel size of $\mathbf{T}^c$, which is 256. $h$ and $w$ denotes the height and width of $\mathbf{T}^c$, respectively. $\mathbf{S}^R$ of $\mathbb{S}$ can also be obtained analogously. Therefore, the context distillation can be expressed as:

$$\mathcal{L}_{CD} = -\tau^2 \sum_{k \in hw \times hw} \sigma(\mathbf{T}_k^R)^{1/\tau} \log \left( \sigma(\mathbf{S}_k^R)^{1/\tau} \right), \tag{5}$$

where $k = (1, 2, ..., hw \times hw)$ is the index item. $\mathbf{T}_k^R$ and $\mathbf{S}_k^R$ denotes the $k$-th item in $\mathbf{T}^R$ and $\mathbf{S}^R$, respectively.

The context distillation presents an efficient solution that does not incur additional inference overhead. As illustrated in Figure 4, our method in (b), which utilizes concatenated features in a whole-to-whole manner, circumvents the potential noise associated with existing context learning methods in (a) for semantic segmentation (Liu et al., 2019; 2024) that rely on layer-to-layer distillation. Our method is specifically tailored to address the potential challenge of incomplete preservation of target region connectivity as shown in Figure 1, rather than solely focusing on the enhancement of feature representations in a generic context.

### 4.4 Overall Loss Function

With the boundary distillation loss $\mathcal{L}_{BD}$ and the context distillation loss $\mathcal{L}_{CD}$, the total loss is expressed as:

$$\mathcal{L} = \mathcal{L}_{SS} + \alpha \mathcal{L}_{BD} + \beta \mathcal{L}_{CD}, \tag{6}$$

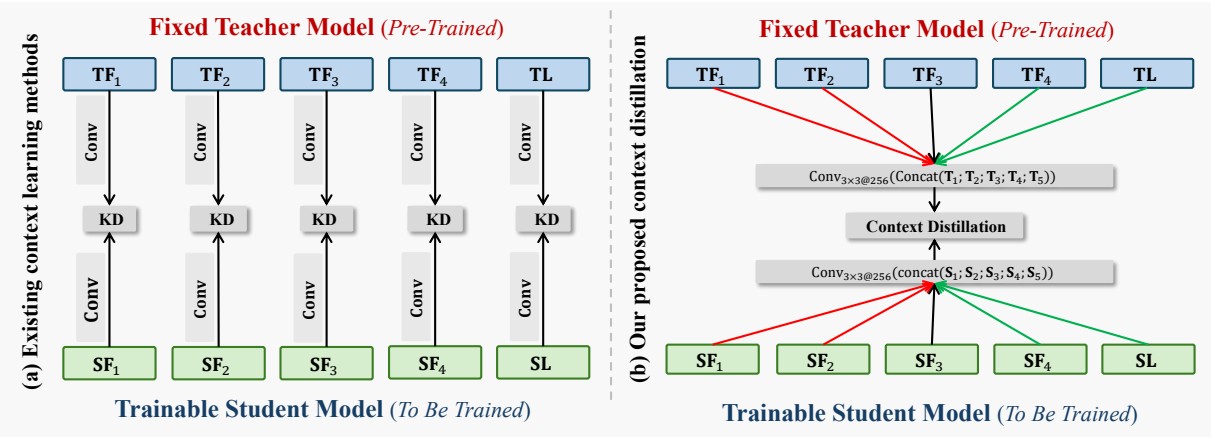

Figure 4: Architecture comparisons between our proposed context distillation (b) and existing context learning methods (a). Our method leverages concatenated features in a whole-to-whole manner, which avoids the potential noise introduced by layer-to-layer distillation modes.

where $\alpha$ and $\beta$ are two weights used to balance different losses. Empirically, these weights have a significant impact on model performance, and unreasonable weight settings may even lead to model collapse in training. To this end, inspired by previous work (Sun et al., 2020; Yang et al., 2023), to enhance the dependence of $\mathbb{S}$ on ground truth labels, we incorporate a weight-decay strategy that promotes a greater focus on $\mathcal{L}_{SS}$ as the training epoch increases. To this end, we initialize a time function as follows:

$$r(t) = 1 - (t-1)/t_{\max}, \tag{7}$$

where $t = (1, 2, ..., t_{\max})$ denotes the current training epoch and $t_{\max}$ is the maximum training epoch. In training, we control the dependence of $\mathcal{L}$ on $\mathcal{L}_{BD}$ and $\mathcal{L}_{CD}$ by using $r(t)$. Therefore, the total loss in Eq. (6) can be formulated as:

$$\mathcal{L} = \mathcal{L}_{SS} + r(t)\alpha\mathcal{L}_{BD} + r(t)\beta\mathcal{L}_{CD}. \tag{8}$$

## 5 Experiments

### 5.1 Datasets and Evaluation Metrics

#### 5.1.1 Datasets

To demonstrate the performance of our method, we conduct experiments on five representative yet challenging datasets: Pascal VOC 2012 Everingham et al. (2010), Cityscapes Cordts et al. (2016), ADE20K Zhou et al. (2017), and COCO-Stuff 10K Caesar et al. (2018) for semantic segmentation (SSeg), as well as MS-COCO 2017 Lin et al. (2014) for instance segmentation (ISeg) and object detection (ODet).

- The Pascal VOC 2012 dataset comprises 20 object classes along with one background class. Following (Zhang et al., 2021; Wang et al., 2020), we utilized the augmented data, resulting in a total of $10,582$ images for *training*, $1,449$ images for *val*, and $1,456$ images for *testing*.

- The Cityscapes comprises a total of $5,000$ finely annotated images, which are partitioned into subsets of $2,975$, $500$, and $1,525$ images designated for *training*, *val*, and *testing*, respectively. In alignment with existing methods (Zheng et al., 2023; Wang et al., 2020; Liu et al., 2019), we exclusively employed the finely labeled data during the training phase to ensure a fair comparison of results.

- The ADE20K dataset has 150 object classes and is organized into three subsets: $20,000$ images for the *training* set, $2,000$ images for the *val* set, and $3,000$ images for the *testing* set.

- The COCO-Stuff 10K dataset is an extension of the MS-COCO dataset Lin et al. (2014), enriched with pixel-wise class labels. It consists of $9,000$ samples designated for *training* and $1,000$ samples allocated for *val*.

- The MS-COCO 2017 dataset comprises 80 object classes and includes a total of $118,000$ images for *training*, and $5,000$ images for *val*.

For data augmentation, random horizontal flip, brightness jittering and random scaling within the range of $[0.5, 2]$ are used in training as in (Wang et al., 2020; Zhang et al., 2021; Liu et al., 2019). Experiments are implemented on the MMRazor framework[2] under the PyTorch platform (Paszke et al., 2019) using 8 NVIDIA GeForce RTX 3090 GPUs. All the inference results are obtained at a single scale.

### 5.1.2 Evaluation metrics

Beyond employing standard metrics, we have also developed two extra specialized evaluation metrics specifically optimized for knowledge distillation on dense image prediction tasks, detailed below:

**Common metrics.** For SSeg, we utilize the mean intersection over union (mIoU) as the primary evaluation metric. For ISeg and ODet, average precision (AP) serves as the principal accuracy-specific metric. To assess model efficiency, we also consider the number of parameters (Params.), Peak GPU memory (Peak GPU Mem.), and the floating-point operations (FLOPs)[3].

**Manifold stability (MFS).** To assess the effectiveness on the learned feature manifolds, we compute the Lipschitz constant ratio between teacher and student models. Specifically, let $f_T^l(x), f_S^l(x) \in \mathbb{R}^{d_l}$ denote the $l$-th layer feature mappings for teacher and student models respectively. The layer-wise Lipschitz constant can be estimated via:

$$L_k^l = \sup_{x \in \mathcal{X}, \|\delta\| \leq \epsilon} \frac{\|f_k^l(x + \delta) - f_k^l(x)\|_2}{\|\delta\|_2}, \quad k \in \{T, S\} \tag{9}$$

where $\mathcal{X}$ is the input space and $\epsilon = 0.1$ controls the perturbation scale. The MFS $\rho_l$ is then computed as:

$$\rho_l = \frac{L_S^l}{L_T^l} \cdot \mathbb{I}(L_T^l > \tau) + \mathbb{I}(L_T^l \leq \tau) \tag{10}$$

with threshold $\tau$ avoiding division by negligible values ($\tau = 0.01$). Values close to 1 indicate well-preserved manifold structure during distillation, while significant deviations suggest potential degradation of geometric properties.

**Local Hausdorff distance (LHD).** For boundary-sensitive tasks like SSeg and ISeg, we introduce a LHD metric to evaluate boundary alignment quality. Specifically, for boundary point sets $\mathcal{B}_p = \{p_i\}_{i=1}^m$ and $\mathcal{B}_g = \{q_j\}_{j=1}^n$, the LHD at point $p_i$ is defined as:

$$\text{LHD}_r(p_i, \mathcal{B}_g) = \min \left\{ \max_{q_j \in N_r(p_i)} d(p_i, q_j), \text{median}\left(\{d(p_i, q_j)\}_{q_j \in N_r(p_i)}\right) \right\}, \tag{11}$$

where the neighborhood $N_r(p_i)$ and final aggregation are defined as:

$$N_r(p_i) = \{q_j \in \mathcal{B}_g \mid \|p_i - q_j\|_2 \leq r\},$$

$$\text{LHD}(\mathcal{B}_p, \mathcal{B}_g) = \frac{1}{|\mathcal{B}_p|} \sum_{i=1}^{|\mathcal{B}_p|} \text{LHD}_r(p_i, \mathcal{B}_g) \cdot \mathbb{I}\left(\text{LHD}_r(p_i, \mathcal{B}_g) \leq \mu + 2\sigma\right), \tag{12}$$

where $r$ is set to 5 in our implementation.

### 5.2 Implementation Details

### 5.2.1 Baselines

For a fair result comparison and considering the realistic resource conditions of edge computing devices (Dong et al., 2023), *for SSeg*, we select **PSPNet-101** (Zhao et al., 2017), **DeepLabv3 Plus-101** (Chen et al.,

---

[2]https://github.com/open-mmlab/mmrazor

[3]While Params/FLOPs are widely used as proxy metrics for model complexity, they do not fully capture the actual performance of models on edge devices. The edge-chip performance remains the ultimate benchmark for EDIP models.

2018), and **Mask2Former** (Cheng et al., 2022) for the teacher models. The student models are compact **PSPNet** and **DeepLabV3+** with ResNet-38, ResNet-18$_{(0.5)}$ and ResNet-18$_{(1.0)}$ (He et al., 2016). Besides, to demonstrate the effectiveness of our method on heterogeneous network architectures, following (Wang et al., 2020; Liu et al., 2019; Yang et al., 2022a), we also employ **MobileNetV2** (Liu, 2018), **EfficientNet-B1** (Tan & Le, 2019), and **SegFormer-B0** (Xie et al., 2021) as the student models. ***For ISeg and ODet***, following (Wang et al., 2024; Zhang & Ma, 2023), we select the representative **GFL** (Li et al., 2020), **Cascade Mask R-CNN** (Cai & Vasconcelos, 2018), and **RetinaNet** (Ross & Dollár, 2017) with ResNet-101/50 and ResNet-50/18 as the teacher model and the student model, respectively. While adopting cutting-edge large foundation models represents the prevailing research trend, we opt for a pragmatic small-model baseline given the currently insurmountable challenges in deploying such large-scale models on edge devices. This application-oriented approach prioritizes practical deployability over model scale, while still providing a meaningful benchmark for edge computing scenarios. During the training and inference phases, aside from our proposed method and specific declarations, all other settings adhere to the configurations outlined in the baseline model.

### 5.2.2  Training details

Following the standard practices as in (Hinton et al., 2015; Liu et al., 2019; Wang et al., 2020), all teacher models are pre-trained on ImageNet-1k by default (Deng et al., 2009), and then fine-tuned on the corresponding dataset before their parameters are fixed in KD. During training, only the student's parameters are updated. As in (Zhang et al., 2021; Phuong & Lampert, 2019), $\tau$ is initialized to 1 and is multiplied by a scaling factor of 1.05 whenever the range of values (across all feature items in a given minibatch) exceeds 0.5. Following (Wang et al., 2020; Sun et al., 2020; Yang et al., 2023), $\alpha$ and $\beta$ is set to 10 and 50, respectively. We fully understand that fine-tuning these base hyperparameters could potentially enhance performance. However, we contend that such adjustments may be redundant and unwarranted.

**For SSeg models**, to accommodate the local hardware limitations, the training images are cropped into a fixed size of $512 \times 512$ pixels as in (Wang et al., 2024; Zhang & Ma, 2023). The SGD is used as the optimizer with the "poly" learning rate strategy. The initial learning rate is set to 0.01, with a power of 0.9. To ensure fairness in the experimental comparisons, the batch size is set to 8 and $t_{\max}$ is set to $40,000$.

**For ISeg and ODet**, the model is trained following the default $1\times$ training schedule, *i.e.*, 12 epochs. The batch size is set to 8, and AdamW is used as the optimizer with the initial learning rate of $1 \times 10^{-4}$ and the weight decay of 0.05. The layer-wise learning rate decay is used and set to 0.9, and the drop path rate is set to 0.4. The given images are resized to the shorter side of 800 pixels, with the longer side not exceeding $1,333$ pixels. In inference, the shorter side of images is consistently set to 800 pixels by default.

### 5.3  Ablation Analysis

In our ablation analysis, we aim to explore answers of the following crucial questions: *1)* the impact of each component within BCKD in Section 5.3.1; *2)* the effectiveness of BCKD across different network architectures in Section 5.3.2; and *3)* the visualized performance and comparisons with other methods in Section 5.3.3. We select the SSeg task as the experimental objective.

### 5.3.1  Effectiveness of each component

To explore answer of the first question, we chose Pascal VOC 2012 Everingham et al. (2010) as the experimental dataset. PSPNet-101 (Zhao et al., 2017) serves as the teacher model, while PSPNet-18$_{(1.0)}$ is used as the student model. Table **??** shows the inference results by adding each component of BCKD into the student model, where we report the experimental results on the *val* set. We can observe that incorporating these components consistently improves the model accuracy, indicating the effectiveness of these components. In particular, adding $\mathcal{L}_{BD}$ resulted in a mIoU↑ gain of 0.91%, which may be attributed to the fact that the object boundary regions are relatively small in proportion to the entire image (Zhang et al., 2021; Liu et al., 2024). With only the implementation of $\mathcal{L}_{BD}$ and $\mathcal{L}_{CD}$, our model can achieve the competitive 73.98% mIoU with 3.20% mIoU↑ improvements, which verifies the importance of boundary and context information in EDIP. Furthermore, this result demonstrates that our$\mathcal{L}_{BD}$ and $\mathcal{L}_{CD}$ do not conflict in practical deployment;

Table 1: Ablation results on Pascal VOC 2012 *val* set (±STD over 3 random seeds). Peak GPU memory (Peak GPU Mem.) measured on NVIDIA RTX 3090.

| Configuration | mIoU (%) | ΔmIoU | Params. | Inf. FLOPs | Tr. FLOPs | Tr. Peak GPU Mem. |
|---|---|---|---|---|---|---|
| Teacher (PSPNet-101) | 77.82±0.15 | - | 70.43M | 411.6G | 411.6G | 24.5GB |
| Student (PSPNet-18) | 70.78±0.21 | - | 15.24M | 106.2G | 106.2G | 8.2GB |
| + $\mathcal{L}_{BD}$ | 71.69±0.18 | +0.91 | 15.24M | 106.2G | 116.8G | 10.7GB |
| + $\mathcal{L}_{CD}$ | 72.55±0.17 | +1.77 | 15.24M | 106.2G | 124.3G | 12.3GB |
| + Both | 73.98±0.14 | +3.20 | 15.24M | 106.2G | 135.6G | 14.1GB |
| + Both + WD | 74.65±0.12 | +3.87 | 15.24M | 106.2G | 142.1G | 15.8GB |

*Note:* STD values calculated over 3 runs with different random seeds (42, 2023, 3407).

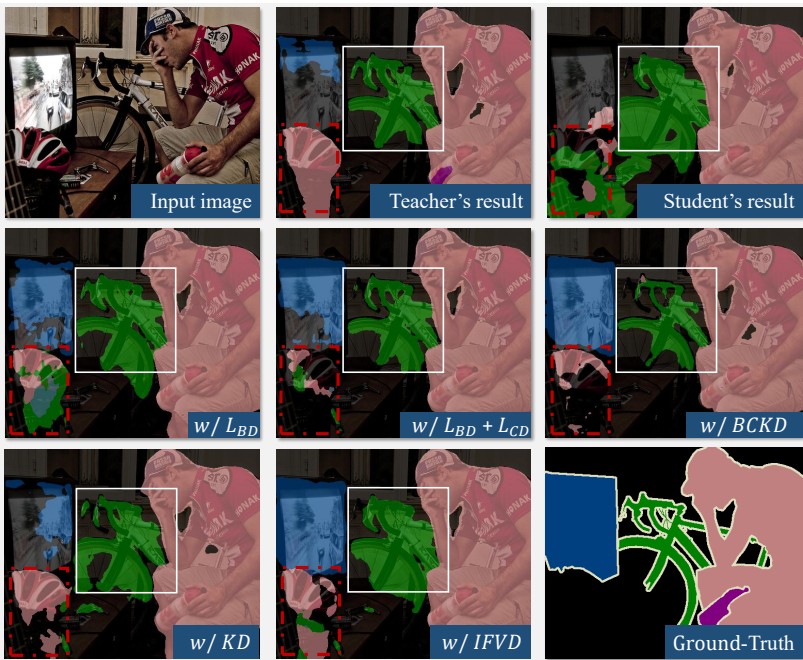

Figure 5: Visualized comparisons and results obtained by adding different components of BCKD. The teacher model and the student model denotes PSPNet-101 (Zhao et al., 2017) and PSPNet-18$_{(1.0)}$ (Zhao et al., 2017), respectively. "*w/*" denotes with the corresponding implementation. Samples are from Pascal VOC 2012 (Everingham et al., 2010).

instead, they complement each other intrinsically to enhance performance. It is also worth noting that compared to the advanced methods in Table 3 that do not utilize ground truth mask, our method also achieves competitive results even without employing the weight-decay strategy. Building upon $\mathcal{L}_{BD}$ and $\mathcal{L}_{CD}$, adding the weight-decay strategy brings 0.57% mIoU↑, which confirms the weight importance of different losses. Furthermore, there is no increase in the number of Params. or FLOPs in the inference stage.

We also employ visualizations as a means of verifying the efficacy of BCKD components incrementally into the baseline model. The obtained results are presented in Figure 5, which demonstrate that the inclusion of $\mathcal{L}_{BD}$ and $\mathcal{L}_{CD}$ in a sequential manner leads to enhanced boundary regions and object connectivity. For example, the "*bike handlebar*". Moreover, the integration of the weight-decay strategy results in further improvement in the overall segmentation predictions. In addition to using white bounding boxes to emphasize the better regions achieved by our method, we also highlighted the regions of prediction failure using red dashed bounding boxes. We can observe that although our BCKD significantly improves the prediction quality of these regions compared to the student model's results, there are still some incomplete predictions. This case may be caused by the spurious correlation between the "*helmet*" and the "*person*" in the used dataset, which can be eliminated through causal intervention (Zhang et al., 2020).

Table 2: Result comparisons on mIoU (%) under different network architectures on the *val* sets of Pascal VOC 2012 (Everingham et al., 2010), Cityscapes (Cordts et al., 2016), and ADE20K (Zhou et al., 2017), and on the *test* set of COCO-Stuff 10K (Caesar et al., 2018). Peak GPU memory in training measured on NVIDIA GeForce RTX 3090. Results include standard deviations from 3 independent runs.

| Methods | Pascal VOC *val* | Params. | Tr. GPU Mem. (GB) |
|---|---|---|---|
| DANet (Fu et al., 2019) | 80.40±0.15 | 83.1**M** | 9.8 |
| $\mathbb{T}$: PSPNet-101 (Zhao et al., 2017) | 77.82±0.12 | 70.4**M** | 8.2 |
| $\mathbb{S}$: PSPNet-38 (Zhao et al., 2017) | 72.65±0.18 | 58.6**M** | 6.1 |
| **+ BCKD$_\mathbf{ours}$** | 75.15±0.14$_{+2.50}$ | 58.6**M** | 6.5 |
| $\mathbb{S}$: EfficientNet-B1 (Tan & Le, 2019) | 69.28±0.25 | 6.7**M** | 3.2 |
| **+ BCKD$_\mathbf{ours}$** | 73.81±0.17$_{+4.53}$ | 6.7**M** | 3.8 |
| $\mathbb{S}$: SegFormer-B0 (Xie et al., 2021) | 66.75±0.22$^‡$ | 3.8**M** | 2.9 |
| **+ BCKD$_\mathbf{ours}$** | 68.88±0.16$_{+2.13}$ | 3.8**M** | 3.3 |

| Methods | Cityscapes *val* | Params. | Tr. GPU Mem. (GB) |
|---|---|---|---|
| HRNet (Sun et al., 2019) | 81.10±0.10 | 65.9**M** | 7.9 |
| $\mathbb{T}$: PSPNet-101 (Zhao et al., 2017) | 78.56±0.11 | 70.4**M** | 8.2 |
| $\mathbb{S}$: PSPNet-38 (Zhao et al., 2017) | 71.26±0.20 | 47.4**M** | 5.8 |
| **+ BCKD$_\mathbf{ours}$** | 73.56±0.15$_{+2.30}$ | 47.4**M** | 6.2 |
| $\mathbb{S}$: EfficientNet-B1 (Tan & Le, 2019) | 60.40±0.30 | 6.7**M** | 3.2 |
| **+ BCKD$_\mathbf{ours}$** | 63.91±0.22$_{+3.51}$ | 6.7**M** | 3.7 |
| $\mathbb{S}$: SegFormer-B0 (Xie et al., 2021) | 76.20±0.13 | 3.8**M** | 2.9 |
| **+ BCKD$_\mathbf{ours}$** | 77.87±0.10$_{+1.67}$ | 3.8**M** | 3.3 |

| Methods | ADE20K *val* | Params. | GPU Mem. (GB) |
|---|---|---|---|
| DeepLab V3 (Chen et al., 2017b) | 43.28±0.18 | 71.3**M** | 8.5 |
| $\mathbb{T}$: PSPNet-101 (Zhao et al., 2017) | 42.19±0.20 | 70.4**M** | 8.2 |
| $\mathbb{S}$: ESPNet (Mehta et al., 2018) | 20.13±0.35 | 0.4**M** | 1.8 |
| **+ BCKD$_\mathbf{ours}$** | 23.65±0.28$_{+3.52}$ | 0.4**M** | 2.1 |
| $\mathbb{S}$: MobileNetV2 (Liu, 2018) | 33.64±0.25 | 8.3**M** | 3.5 |
| **+ BCKD$_\mathbf{ours}$** | 36.59±0.19$_{+2.95}$ | 8.3**M** | 4.0 |
| $\mathbb{S}$: SegFormer-B0 (Xie et al., 2021) | 37.40±0.22 | 3.8**M** | 2.9 |
| **+ BCKD$_\mathbf{ours}$** | 38.75±0.18$_{+1.35}$ | 3.8**M** | 3.3 |
| $\mathbb{T}$: Mask2Former (Cheng et al., 2022) | 47.20±0.16 | 44.0**M** | 6.8 |
| $\mathbb{S}$: **ESPNet + BCKD$_\mathbf{ours}$** | 24.26±0.30$_{+4.13}$ | 0.4**M** | 2.3 |
| $\mathbb{S}$: **MobileNetV2 + BCKD$_\mathbf{ours}$** | 37.09±0.21$_{+3.45}$ | 8.3**M** | 4.2 |
| $\mathbb{S}$: **SegFormer-B0 + BCKD$_\mathbf{ours}$** | 39.61±0.17$_{+2.21}$ | 3.8**M** | 3.5 |

| Methods | COCO 10K *test* | FLOPs | GPU Mem. (GB) |
|---|---|---|---|
| SegVIT (Zhang et al., 2022a) | 50.3±0.15 | 383.9**G** | 11.2 |
| $\mathbb{T}$: DeepLabV3 Plus-101 (Chen et al., 2018) | 33.10±0.22 | 366.9**G** | 9.1 |
| $\mathbb{S}$: MobileNetV2 (Liu, 2018) | 26.29±0.28 | 1.4**G** | 3.5 |
| **+ BCKD$_\mathbf{ours}$** | 28.31±0.23$_{+2.02}$ | 1.4**G** | 4.0 |
| $\mathbb{S}$: SegFormatter-B0 (Xie et al., 2021) | 35.60±0.18 | 8.4**G** | 2.9 |
| **+ BCKD$_\mathbf{ours}$** | 35.92±0.15$_{+0.32}$ | 8.4**G** | 3.3 |

### 5.3.2 Effectiveness across network architectures

In this section, we evaluate the effectiveness of our BCKD across various network architectures for SSeg using: Pascal VOC 2012 (Everingham et al., 2010), Cityscapes (Cordts et al., 2016), ADE20K (Zhou et al., 2017), and COCO-Stuff 10K (Caesar et al., 2018). The obtained results under different network architectures are given in Table 2. For the purpose of comparing experimental results, we also include the results of a large model

Table 3: Comparisons on mIoU (%) with state-of-the-art methods on the *val* sets of Pascal VOC 2012 (Everingham et al., 2010) and Cityscapes (Cordts et al., 2016), and ADE20K (Zhou et al., 2017). "‡" denotes our re-implemented result based on the released codes due to inconsistencies in experimental settings. "KD Manner": knowledge distillation manner, which contains of layer-to-layer (L2L) and whole-to-whole (W2W) as illustrated in Figure 4. "GTM': ground-truth mask used to obtain the pre-extraction image boundaries. "Bod." denotes the boundary type, and "Cont." denotes the context type. "*E*": physical edge. "*B*": semantic boundary. "*P*": pixel-wise. "*I*": image-wise. "*O*": object-wise.

| Methods | KD Manner | GTM? | Bod. | Cont. | Pascal VOC 2012 | Cityscapes | ADE20K |
|---|---|---|---|---|---|---|---|
| $\mathbb{T}$: PSPNet-101 (Zhao et al., 2017) | | | | | 77.82% | 78.56% | 42.19% |
| $\mathbb{S}$: PSPNet-18$_{(1.0)}$ (Zhao et al., 2017) | | | | | 70.78% | 69.10% | 33.82% |
| + KD (Hinton et al., 2015) | L2L | ✗ | ✗ | ✗ | 71.28$^{\ddagger}_{+0.50}$ | 71.20$_{+2.10}$ | 34.33$^{\ddagger}_{+0.51}$ |
| + SKD (Liu et al., 2019) | L2L | ✓ | $E$ | $P$ | 73.05$_{+2.27}$ | 71.45$_{+2.35}$ | 34.65$_{+0.83}$ |
| + SCKD (Zhu & Wang, 2021) | L2L | ✗ | ✗ | ✗ | 72.33$_{+1.55}$ | 72.10$_{+3.00}$ | 34.76$_{+0.94}$ |
| + CIRKD (Yang et al., 2022a) | L2L | ✗ | ✗ | $I$ | 73.57$^{\ddagger}_{+2.79}$ | 72.25$_{+3.15}$ | 34.93$_{+1.11}$ |
| + IFD (Chen et al., 2022) | L2L | ✗ | ✗ | ✗ | 73.88$^{\ddagger}_{+3.10}$ | 72.63$_{+3.53}$ | 35.15$^{\ddagger}_{+1.33}$ |
| + FGKD (Yang et al., 2022b) | L2L | ✗ | ✗ | $O$ | 72.90$^{\ddagger}_{+2.12}$ | 72.55$_{+3.45}$ | 35.24$_{+1.42}$ |
| + CWT (Liu et al., 2023) | L2L | ✗ | ✗ | $O$ | 73.06$^{\ddagger}_{+2.28}$ | 72.60$_{+3.50}$ | 35.21$^{\ddagger}_{+1.39}$ |
| + SlimSeg (Xue et al., 2022) | L2L | ✓ | $E$ | ✗ | 74.08$_{+3.30}$ | 73.95$_{+4.85}$ | 37.12$_{+3.30}$ |
| + BGLSSeg (Zhou et al., 2024) | L2L | ✓ | $E$ | ✗ | 74.27$^{\ddagger}_{+3.49}$ | 74.10$^{\ddagger}_{+5.00}$ | 36.49$^{\ddagger}_{+2.67}$ |
| + FAM (Pham et al., 2024) | L2L | ✗ | ✗ | ✗ | 74.28$^{\ddagger}_{+3.50}$ | 74.25$_{+5.15}$ | 36.82$^{\ddagger}_{+3.00}$ |
| + CrossKD (Wang et al., 2024) | L2L | ✗ | ✗ | ✗ | 74.28$^{\ddagger}_{+3.50}$ | 74.28$_{+5.18}$ | 36.72$^{\ddagger}_{+2.90}$ |
| + BPKD (Liu et al., 2024) | L2L | ✓ | $E$ | ✗ | 74.30$^{\ddagger}_{+3.52}$ | 74.29$_{+5.19}$ | 37.07$^{\ddagger}_{+3.25}$ |
| **+ BCKD$_{\mathbf{ours}}$** | W2W | ✗ | $B$ | $P$ | 74.65$_{+3.87}$ | 74.92$_{+5.82}$ | 37.62$_{+3.80}$ |
| + IFVD (Wang et al., 2020) | L2L | ✗ | ✗ | ✗ | 74.05$_{+3.27}$ | 74.41$_{+5.31}$ | 36.63$^{\ddagger}_{+3.35}$ |
| **+ IFVD + BCKD$_{\mathbf{ours}}$** | W2W | ✗ | $B$ | $P$ | 74.82$_{+4.04}$ | 74.99$_{+5.89}$ | 37.73$_{+3.91}$ |
| + TAT (Lin et al., 2022) | L2L | ✗ | ✗ | $O$ | 74.02$^{\ddagger}_{+3.24}$ | 74.48$_{+5.38}$ | 37.12$_{+3.30}$ |
| **+ TAT + BCKD$_{\mathbf{ours}}$** | W2W | ✗ | $B$ | $P\&O$ | 74.71$_{+3.93}$ | 74.97$_{+5.87}$ | 38.12$_{+4.30}$ |
| + SSTKD (Ji et al., 2022) | L2L | ✓ | $E$ | ✗ | 73.91$_{+3.13}$ | 74.60$_{+5.50}$ | 37.22$_{+3.40}$ |
| **+ SSTKD + BCKD$_{\mathbf{ours}}$** | W2W | ✓ | $B\&E$ | $P$ | 74.52$_{+3.74}$ | 74.90$_{+5.80}$ | 38.24$_{+4.52}$ |

that does not utilize knowledge distillation for each dataset. From this table, we can observe that deploying BCKD on different network architectures can lead to continuous performance improvements. For example, when employing PSPNet-38 (Zhao et al., 2017), EfficientNet-B1 (Tan & Le, 2019), and SegFormer-B0 (Xie et al., 2021) as student models while keeping the teacher model PSPNet-101 (Zhao et al., 2017) unchanged, our BCKD achieves mIoU↑ improvements of 2.50%, 4.53%, and 2.13% on Pascal VOC *val*, and 2.30%, 3.51%, and 1.67% on Cityscapes *val*, respectively. The performance gains demonstrate the effectiveness of our method not only within the same network architecture but also across network architectures, indicating sustained performance enhancements. This phenomenon also highlights the generalization capacity of our method. Besides, similar conclusions can be also drawn from our experimental results on ADE20K *val* and COCO-Stuff 10K *test* sets. Deploying different teacher models on the ADE20K *val* dataset are also conducted, where Mask2Former (Cheng et al., 2022) is utilized as the teacher model, and ESPNet (Mehta et al., 2018), MobileNetV2 (Liu, 2018), and SegFormer-B0 are used as the student models. The results demonstrate that our method yields mIoU↑ improvements of 4.13%, 3.45%, and 2.21% on ESPNet, MobileNetV2, and SegFormer-B0 (Xie et al., 2021), respectively, showcasing the strong flexibility. On efficiency, our method leverages the KD framework, resulting in no increase in Params. or FLOPs. Consequently, we achieve both improved accuracy and fast inference speed.

Table 4: Comparisons on mIoU (%), MFS and LHD with state-of-the-art methods on the *test* set of COCO-Stuff 10K (Caesar et al., 2018). Standard deviations are computed from 5 independent runs.

| | | | |
|---|---|---|---|
| $\mathbb{T}$: DeepLabV3 Plus-101 | 33.10±0.15% | 1.00±0.00 | 0.00±0.00 |
| $\mathbb{S}$: DeepLabV3 Plus-18 | 26.33±0.18% | 0.45±0.08 | 4.21±0.35 |
| Methods | mIoU (%) | MFS ($\rho_l$) | LHD |
| + KD (Hinton et al., 2015) | 27.21±0.16$_{+0.88}$ | 0.63±0.07 | 3.82±0.41 |
| + SKD (Liu et al., 2019) | 27.27±0.15$_{+0.94}$ | 0.67±0.06 | 3.77±0.38 |
| + SCKD (Zhu & Wang, 2021) | 27.38±0.14$_{+1.05}$ | 0.72±0.08 | 3.63±0.42 |
| + CIRKD (Yang et al., 2022a) | 27.68±0.13$_{+1.35}$ | 0.81±0.05 | 3.45±0.33 |
| + IFD (Chen et al., 2022) | 27.91±0.12$_{+1.58}$ | 0.88±0.04 | 3.28±0.29 |
| + FGKD (Yang et al., 2022b) | 28.00±0.11$_{+1.67}$ | 0.91±0.03 | 3.12±0.31 |
| + CWT (Liu et al., 2023) | 28.18±0.10$_{+1.85}$ | 0.95±0.02 | 2.98±0.22 |
| + IFVD (Wang et al., 2020) | 28.35±0.09$_{+2.02}$ | 1.02±0.03 | 2.77±0.25 |
| + C2VKD (Zheng et al., 2023) | 28.42±0.08$_{+2.09}$ | 1.08±0.04 | 2.63±0.24 |
| + FAM (Pham et al., 2024) | 28.48±0.08$_{+2.15}$ | 1.12±0.03 | 2.55±0.21 |
| + CrossKD (Wang et al., 2024) | 28.53±0.07$_{+2.20}$ | 1.15±0.02 | 2.48±0.19 |
| + SSTKD (Ji et al., 2022) | 28.70±0.06$_{+2.37}$ | 1.23±0.03 | 2.31±0.17 |
| **+ BCKD** | 29.22±0.05$_{+2.89}$ | 1.41±0.02 | 1.89±0.15 |
| + TAT (Lin et al., 2022) | 28.74±0.06$_{+2.41}$ | 1.25±0.04 | 2.25±0.18 |
| **+ TAT + BCKD** | 29.29±0.04$_{+3.11}$ | 1.44±0.01 | 1.82±0.14 |
| + SlimSeg (Xue et al., 2022) | 28.50±0.08$_{+2.17}$ | 1.13±0.03 | 2.52±0.20 |
| **+ SlimSeg + BCKD** | 29.45±0.04$_{+3.12}$ | 1.46±0.01 | 1.79±0.13 |
| + BPKD (Liu et al., 2024) | 28.66±0.07$_{+2.33}$ | 1.21±0.02 | 2.36±0.16 |
| **+ BPKD + BCKD** | 29.60±0.03$_{+3.27}$ | 1.49±0.01 | 1.72±0.12 |

### 5.3.3 Visualized comparisons

The visualized comparisons on the SSeg task with the baseline teacher and student models, and large models without the KD strategy are given in Figure 6. As highlighted by the white bounding boxes, the obtained results on Pascal VOC 2012 (Everingham et al., 2010), Cityscapes (Cordts et al., 2016), ADE20K (Zhou et al., 2017), and COCO-Stuff 10K (Caesar et al., 2018) demonstrate that BCKD yields significant improvements on both the boundary region completeness and the target region connectivity, when compared with the small student model's results. For example, the "*cow*" in Pascal VOC 2012, the "*guidepost*" in Cityscapes, the "*desk*" and the "*TV bench*" in ADE20K, the "*bus*", the "*tennis racket*", and the "*guideboard*" in COCO-Stuff 10K. The results obtained are basically the same as those of the large teacher model. Besides, compared to large models with higher model complexity (*i.e.*, DANet (Fu et al., 2019) and HRNet (Sun et al., 2019)) on Pascal VOC 2012 and Cityscapes, although our method is not as competitive as theirs on quantitative results, our method achieves better predictions on object boundaries and small objects, which validate the effectiveness and emphasize the importance of boundary distillation and context distillation. With the help of our method, the student model is also able to predict better masks for certain fine-grained objects. For example, the "*cow's ear*" and the "*person's leg*".

### 5.4 Comparisons With SOTA Methods on SSeg

In this section, we explore the accuracy and the effectiveness of the joint implementation of BCKD with the state-of-the-art (SOTA) KD methods on SSeg. To ensure a fair comparison, PSPNet-101 (Zhao et al., 2017) and DeepLabV3 Plus-101 (Chen et al., 2018) are employed as the teacher models, while PSPNet-18$_{(1.0)}$ and DeepLabV3 Plus-18 (Chen et al., 2018) serve as the student models. The specific settings for each student model are described in detail in the provided table. Some results are re-implemented by us on the released code due to inconsistencies in experimental settings and are marked with "‡" in the given tables.

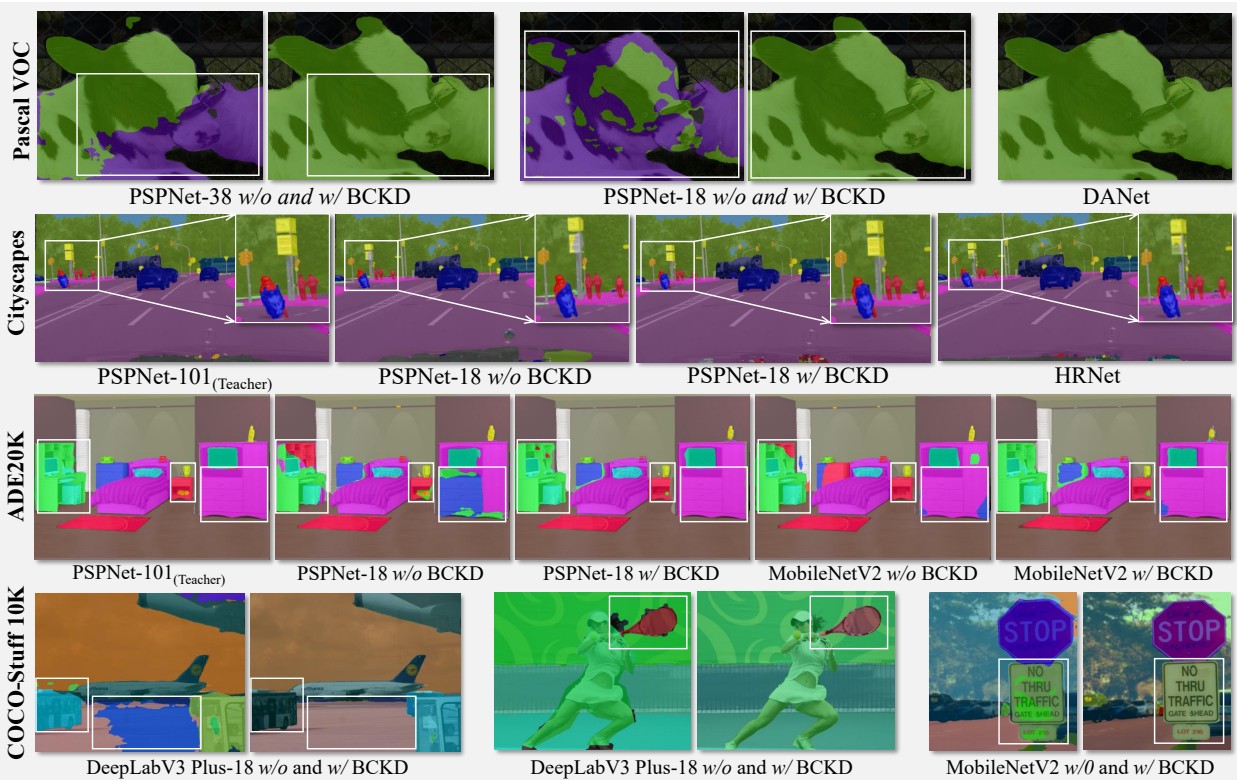

Figure 6: Visualizations on SSeg. DANet (Fu et al., 2019) and HRNet (Sun et al., 2019) have been included for comparison as well. "*w/o*" means "without" and "*w/*" means "with", indicating whether our method is NOT implemented or implemented. The white bounding boxes highlight the regions where our method predicts better.

### 5.4.1 Superiority of BCKD

Compared to the SOTA KD methods on SSeg, on the top half of Table 3 and Table 4, we can observe that our BCKD can surpass these methods. BCKD boosts the student model by 3.87%, 5.82%, and 3.80% mIoU↑ on the *val* sets of Pascal VOC 2012 (Everingham et al., 2010), Cityscapes (Cordts et al., 2016), and ADE20K (Zhou et al., 2017), respectively. Compared to the current SOTA KD methods on these datasets, BCKD outperforms IFVD (Wang et al., 2020), TAT (Lin et al., 2022), and SSTKD (Ji et al., 2022) on Pascal VOC 2012 by 0.6%, 0.63%, and 0.74% mIoU↑, respectively. The visualized comparison results with the classic KD (Hinton et al., 2015) and SOTA IFVD methods are presented in the last row of Figure 5. It can be observed that our method demonstrates significant advantages in capturing the connectivity of small objects as well as the integrity of the boundary masks. Furthermore, BCKD achieves higher mIoU than SOTA methods on Cityscapes and ADE20K datasets as well. On the COCO-Stuff 10K (Caesar et al., 2018) datasets in Table 4, our method surpasses the student model and the SOTA TAT model by 2.89% and 0.48% mIoU↑, respectively. As demonstrated in Table 4, the proposed BCKD framework also exhibits significant advantages in terms of MFS ($\rho_l$) and LHD. These results not only indicate an overall performance improvement but also validate the effectiveness of our novel boundary distillation and context distillation, which were designed to address these critical aspects of the learning process. Since inference is only conducted on the student model, our method does not introduce any increase in model complexity. These results across different datasets can confirm that the task-specific knowledge is indeed more effective in practice compared to general knowledge.

### 5.4.2 Effectiveness of the joint implementation

The results on the joint implementation of BCKD and SOTA KD methods are presented on the lower half of Table 3 and Table 4, respectively. It can be observed that, on top of BCKD, further adding IFVD (Wang et al., 2020), TAT (Lin et al., 2022), and SSTKD (Ji et al., 2022) yields consistent performance gains, with

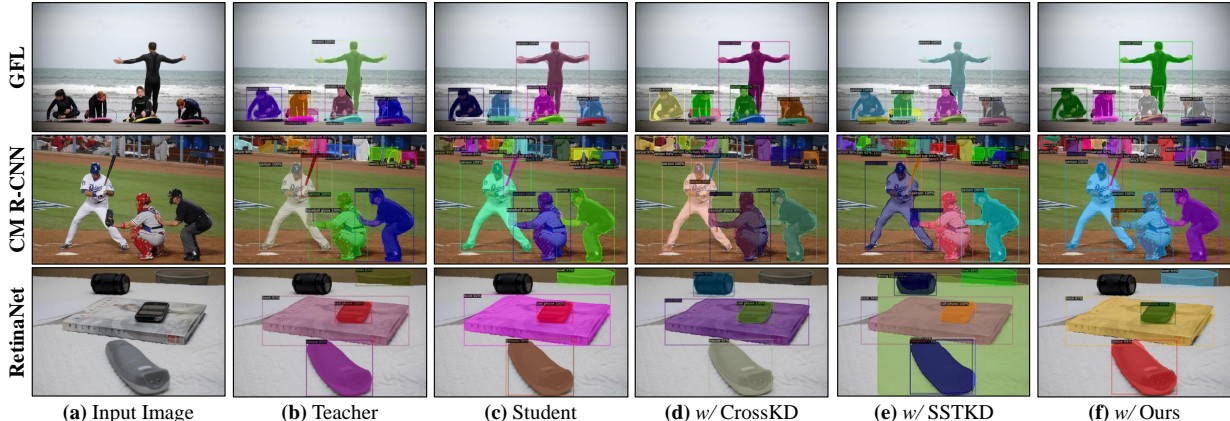

|  | (a) Input Image | (b) Teacher | (c) Student | (d) w/ CrossKD | (e) w/ SSTKD | (f) w/ Ours |

Figure 7: Visualization results on ISeg and ODet. "*w/*" means "with", indicating that the corresponding knowledge distillation method is deployed based on the student model. We chose the state-of-the-art methods CrossKD Wang et al. (2024) and SSTKD Ji et al. (2022) for comparison.

mIoU↑ improvements of 0.77%, 0.69%, and 0.61% on the *val* set of Pascal VOC 2012, respectively. This can be attributed to the fact that BCKD contains semantic boundary and context that is not present in these methods, further demonstrating the importance of semantic boundary and context information for the SSeg task. However, adding SSTKD (Ji et al., 2022) on top of BCKD resulted in a performance decrease (*i.e.*, 0.13% mIoU↓ on Pascal VOC 2012 and 0.02% mIoU↓ on Cityscapes) compared to the accuracy on BCKD. We guess that this may be because SSTKD uses superficial image texture information, which is non-semantic and contain some noise relative to the extracted semantic boundaries. On COCO-Stuff 10K, we can observe that our method further enhances the performance of all SOTA methods, including TAT (Lin et al., 2022), SlimSeg (Xue et al., 2022), and BPKD (Liu et al., 2024), and finally achieves 29.60% mIoU on the *test* set.

## 5.5 Comparisons With SOTA Methods on ISeg and ODet

The quantitative result comparisons on ISeg and ODet are presented in Table 5. The obtained results indicate that our method can consistently outperform existing methods across various baseline models, demonstrating its strong generalization and versatility. Specifically, we achieve AP scores of 35.8%/38.8%, 37.0%/42.5%, and 33.3%/38.5% for instance segmentation masks (*i.e.*, $AP^m$) and object detection bounding boxes (*i.e.*, $AP^b$) on the GFL-18 (Li et al., 2020), Cascade Mask R-CNN-50 (Cai & Vasconcelos, 2018), and RetinaNet-50 (Ross & Dollár, 2017), respectively. In comparison with the SOTA CrossKD (Wang et al., 2024) and SSTKD (Ji et al., 2022), our method demonstrates an average performance improvement of approximately 0.5%. This enhancement serves to validate the effectiveness of our proposed method. The results also demonstrate significant advantages in both MFS and LHD, indicating its capability in preserving the structure of learned feature manifolds while maintaining high precision in boundary-sensitive tasks. These results substantiate the effectiveness of our boundary and context distillation in maintaining geometric consistency and minimizing alignment errors.

The visual comparison results with baseline methods and SOTA methods are shown in Figure 7. It is observed that, relative to the baseline student models, the application of various KD strategies enhances the prediction results for specific classes (*e.g.*, the *person*", the *book*", and the *surfboard*"), thereby affirming the effectiveness of KD in dense image prediction tasks. Moreover, when compared to the SOTA methods CrossKD and SSTKD, our method demonstrates improved connectivity in object regions and boundary integrity (*e.g.*, the *person*" and the *baseball bat*"), highlighting the effectiveness of our proposed context distillation and boundary distillation strategies tailored for the targeted tasks. Additionally, our method addresses the issue of overlapping predicted bounding boxes (*e.g.*, the *mouse*" and the "*chair*"), a benefit attributed to the enriched contextual information incorporated into the student model via context distillation.

Furthermore, we also observed a significant phenomenon wherein our method effectively reduces the occurrence of hallucinations in the student model's predictions. Specifically, as depicted in the last column of Figure 7,

Table 5: Result comparisons with the state-of-the-art methods on the *val* set of MS-COCO 2017 Lin et al. (2014) for ISeg and ODet. "CM R-CNN": Cascade Mask R-CNN. mAP$^m$ and mAP$^b$ denotes the average precision on instance segmentation mask and object detection bounding box, respectively.

| Methods | AP$^m$ (%) | AP$^b$ (%) | FPS | MFS ($\rho_l$) | LHD |
|---|---|---|---|---|---|
| $\mathbb{T}$: GFL-50 (Li et al., 2020) | 36.8 | 40.2 | 19.4 | 1.00 | 0.00 |
| $\mathbb{S}$: GFL-18 (Li et al., 2020) | 33.1 | 35.8 | 23.7 | 0.82 | 3.20 |
| + FGD (Yang et al., 2022b) | 34.0 | 36.6 | 23.7 | 0.85 | 2.90 |
| + SKD (Liu et al., 2019) | 34.3 | 36.9 | 23.7 | 0.86 | 2.70 |
| + GID (Dai et al., 2021) | 34.6 | 37.8 | 23.7 | 0.89 | 2.50 |
| + LD (Zheng et al., 2022) | 34.8 | 38.0 | 23.7 | 0.90 | 2.30 |
| + PKD (Cao et al., 2022b) | 35.0 | 38.0 | 23.7 | 0.91 | 2.20 |
| + CrossKD (Wang et al., 2024) | 35.3 | 38.1 | 23.7 | 0.92 | 1.90 |
| + SSTKD (Ji et al., 2022) | 35.2 | 38.3 | 23.7 | 0.93 | 1.80 |
| + BCKD$_{ours}$ | **35.8** | **38.8** | 23.7 | **0.95** | **1.60** |
| $\mathbb{T}$: CM R-CNN-101 (Cai & Vasconcelos, 2018) | 37.3 | 42.9 | 13.1 | 1.00 | 0.00 |
| $\mathbb{S}$: CM R-CNN-50 (Cai & Vasconcelos, 2018) | 36.5 | 41.9 | 16.1 | 0.88 | 2.10 |
| + FGD (Yang et al., 2022b) | 35.3 | 42.1 | 16.1 | 0.87 | 2.20 |
| + SKD (Liu et al., 2019) | 36.5 | 42.2 | 16.1 | 0.89 | 2.00 |
| + GID (Dai et al., 2021) | 36.7 | 42.0 | 16.1 | 0.90 | 1.90 |
| + LD (Zheng et al., 2022) | 36.8 | 42.1 | 16.1 | 0.91 | 1.80 |
| + PKD (Cao et al., 2022b) | 36.8 | 42.0 | 16.1 | 0.92 | 1.70 |
| + CrossKD (Wang et al., 2024) | 36.9 | 42.2 | 16.1 | 0.93 | 1.60 |
| + SSTKD (Ji et al., 2022) | 37.0 | 42.2 | 16.1 | 0.94 | 1.50 |
| + BCKD$_{ours}$ | **37.0** | **42.5** | 16.1 | **0.96** | **1.40** |
| $\mathbb{T}$: RetinaNet-101 (Ross & Dollár, 2017) | 33.5 | 38.9 | 13.5 | 1.00 | 0.00 |
| $\mathbb{S}$: RetinaNet-50 (Ross & Dollár, 2017) | 31.7 | 37.4 | 17.7 | 0.85 | 2.40 |
| + FGD (Yang et al., 2022b) | 32.1 | 37.7 | 17.7 | 0.86 | 2.20 |
| + SKD (Liu et al., 2019) | 32.5 | 37.5 | 17.7 | 0.87 | 2.12 |
| + GID (Dai et al., 2021) | 32.8 | 37.6 | 17.7 | 0.88 | 2.03 |
| + LD (Zheng et al., 2022) | 33.1 | 37.8 | 17.7 | 0.89 | 1.90 |
| + PKD (Cao et al., 2022b) | 33.0 | 37.8 | 17.7 | 0.90 | 1.88 |
| + CrossKD (Wang et al., 2024) | 33.2 | 38.0 | 17.7 | 0.91 | 1.75 |
| + SSTKD (Ji et al., 2022) | 33.1 | 38.1 | 17.7 | 0.92 | 1.66 |
| + BCKD$_{ours}$ | **33.3** | **38.5** | 17.7 | **0.94** | **1.54** |

both the teacher and student models fail to identify the "*camera*", while the CrossKD and SSTKD methods mistakenly classify the "*camera*" as the "*bottle*". In contrast, our approach accurately recognizes the "*camera*" as a background object, aligning with definitions. We hypothesize that this discrepancy may stem from the confusion of target knowledge caused by task-irrelevant KD during the training process. Our proposed task-specific BCKD is inherently designed to alleviate such confusion from the outset.

## 6 Conclusion and Future Work

In this work, we propose a customized boundary and context knowledge distillation (BCKD) method tailored for efficient dense image prediction tasks on AI accelerator, including semantic segmentation, instance segmentation, and object detection. Our approach significantly narrows the performance gap between compact, efficient models and their larger, more accurate counterparts while maintaining computational efficiency. Specifically, BCKD enhances boundary-region completeness and ensures object-region connectivity, leading

to consistent accuracy improvements across diverse challenging benchmarks and architectures. Theoretical analysis further corroborates the effectiveness of our method.

As a generalizable method, in the future, we plan to extend BCKD to additional dense visual tasks (*e.g.*, pose estimation and image generation) and investigate its adaptation to emerging architectures (*e.g.*, Vision Mamba) to better support model compression for AI accelerator deployment. Besides, we will explore synergies between BCKD and large foundation models (*e.g.*, Segment Anything Model and vision-language models) to further enhance the robustness of lightweight dense predictors under adverse conditions. Furthermore, investigating BCKD in next-generation models, such as Vision Mamba (Vim), also represents a highly practical and instructive research direction.

## Acknowledgements

This work was supported by the National Natural Science Foundation of China Young Scholar Fund Category B (62522216), Young Scholar Fund Category C (62402408), the HKSAR RGC General Research Fund (16219025), HKSAR RGC Early Career Scheme (26208924), and HKSAR RGC General Research Fund (16208823).

## Broader Impact Statement

We acknowledge the broader implications of deploying EDIP models in real-world applications. While EDIP models are designed to be efficient and effective for edge device deployment, there are potential risks associated with their use in high-stakes scenarios such as mass surveillance, facial recognition, or other security-related applications. These applications may raise ethical concerns and require careful adherence to regulatory frameworks to ensure responsible use. As such, we emphasize the importance of ethical considerations and legal compliance when deploying such models, and recommend the implementation of safeguards to protect user privacy.

We also recognize that biases present in the training data may be inadvertently amplified during model distillation or other knowledge transfer processes in EDIP systems. For example, the "helmet/person" correlation serves as a case study demonstrating such dataset bias. To address this, we will propose methods such as fairness-aware training and post-hoc bias correction to reduce the likelihood of biased outcomes and ensure more equitable model behavior.

Finally, we also consider the environmental impact of our model training process, which may involve up to 40,000 training epochs. Although our model is efficient in inference, the high computational and energy demands during training raise concerns about sustainability. To mitigate this, we suggest future strategies such as progressive training, knowledge distillation with low-rank adaptation, and early stopping techniques, where applicable, to reduce the overall environmental footprint of model development.

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

# A  Appendix

**Theoretical Analysis**

BCKD is theoretically grounded in differential geometry and spectral graph theory. In the Appendix, we will analyze its components through measurable properties of the learned feature manifolds $\mathcal{M}_T$ (teacher) and $\mathcal{M}_S$ (student), with proofs connecting to the empirical results in Section 5.3.

## A.1  Boundary-Aware Manifold Alignment

The proposed $\mathcal{L}_{BD}$ in Eq. (3) induces geometric consistency between teacher and student decision boundaries.

**Theorem 1** (Boundary Consistency). *Under $\mathcal{L}_{BD}$ minimization with $\tau > 1$, for any boundary point $x_b$, we have:*

$$\|\mathbf{J}_T(x_b) - \mathbf{J}_S(x_b)\|_F \leq \sqrt{2\mathcal{L}_{BD}/\tau^2} + \mathcal{O}(e^{-\tau}), \tag{13}$$

*where $\mathbf{J}.$ are Jacobian matrices of the feature maps.*

*Proof.* The temperature-scaled gradients satisfy:

$$\nabla\mathcal{L}_{BD} = \mathbb{E}_{x_b}\left[\frac{\sigma_T^{1/\tau}}{\sigma_S^{1/\tau}}\nabla\log\sigma_S\right], \tag{14}$$

$$\|\nabla L_T - \nabla L_S\|^2 \leq 2(1 - \cos\theta), \tag{15}$$

where $\theta$ is the angle between $\mathbb{S}$ and $\mathbb{T}$ gradients. Applying Taylor expansion (Kanwal & Liu, 1989) at high $\tau$, then we can obtain:

$$\cos\theta \geq 1 - \frac{1}{2}\mathcal{L}_{BD}\tau^{-2} + \mathcal{O}(\tau^{-4}). \tag{16}$$

The Jacobian bound follows from Pinsker's inequality applied to the manifold tangent spaces. □

This theoretical guarantee explains the 0.91% mIoU improvement observed in Table **??**, as aligned Jacobians ensure consistent boundary localization.

## A.2  Contextual Graph Preservation

The proposed $\mathcal{L}_{CD}$ in Eq. (5) maintains spectral properties critical for dense image prediction tasks:

**Theorem 2** (Spectral Convergence). *For eigenvalues $\{\lambda_i\}$ of relation matrices $\mathbf{T}^R, \mathbf{S}^R$, we have:*

$$\max_i |\lambda_i^T - \lambda_i^S| \leq \|\mathbf{T}^R - \mathbf{S}^R\|_F \leq \sqrt{d}\mathcal{L}_{CD}. \tag{17}$$

*Proof.* Applying *Weyl's inequality* for symmetric matrices, we can obtain:

$$|\lambda_i^T - \lambda_i^S| \leq \|\Delta\mathbf{R}\|_2 \leq \|\Delta\mathbf{R}\|_F, \tag{18}$$

where the heat kernel continuity follows from:

$$\|e^{-\tau\mathcal{L}_T} - e^{-\tau\mathcal{L}_S}\|_F \leq \tau\sup_{t\in[0,\tau]}\|e^{-t\mathcal{L}_T}(\mathcal{L}_T - \mathcal{L}_S)e^{-(\tau-t)\mathcal{L}_S}\| \leq \tau e^{\tau\|\mathcal{L}_T\|}\mathcal{L}_{CD} \tag{19}$$

□

As shown in Table **??**, compared with the baseline model, the 1.77% mIoU gain directly reflects this eigenvalue stability.

### A.3 Multi-Scale Geometric Consistency

As illustrated in Figure 4, the feature concatenation and projection operation in Section 4.1 preserves topological invariants:

**Proposition 1** (Topological Preservation). *The mapping $\phi : \prod_i \mathcal{M}_T^{(i)} \to \mathcal{M}_T^{concat}$ satisfies:*

$$\beta_k(\mathcal{M}_T^{concat}) = \sum_{i=1}^{5} \beta_k(\mathcal{M}_T^{(i)}), \quad k = 0, 1, 2 \tag{20}$$

*where $\beta_k$ are Betti numbers.*

*Proof.* The 3×3 convolution operation is a diffeomorphism, thus we have:

$$\beta_k(\phi(\mathbf{T}^c)) = \beta_k(\mathbf{T}^c) \quad \text{(invariance)} \tag{21}$$

$$= \beta_k(\oplus_i \mathcal{M}_T^{(i)}) = \sum_{i=1}^{5} \beta_k(\mathcal{M}_T^{(i)}) \quad \text{(Künneth formula)}$$

The dimensionality bound follows from the classical projection theorem as in (Falconer & Howroyd, 1996). □

### A.4 Training Dynamics Interpretation

The used weight decay in Eq. (8) induces phased learning:

**Theorem 3** (Annealed Convergence). *With $r(t) = 1 - (t-1)/t_{\max}$ and Robbins-Monro conditions on learning rate $\eta_t$:*

$$\lim_{t \to t_{\max}} \mathbb{P}(\mathcal{L} = \mathcal{L}_{SS}) = 1 \tag{22}$$

*Proof.* Decompose the gradient flow:

$$\frac{d\mathcal{L}}{dt} = -\eta_t \|\nabla \mathcal{L}_{SS}\|^2 - \eta_t r(t)^2 (\alpha^2 \|\nabla \mathcal{L}_{BD}\|^2 + \beta^2 \|\nabla \mathcal{L}_{CD}\|^2) \tag{23}$$

As $r(t) \to 0$, the right terms vanish asymptotically. The convergence follows the stochastic approximation theory (Lai, 2003). □

**Pseudo-Code of BCKD**

## B BCKD Algorithm

The Boundary and Context Knowledge Distillation (BCKD) consists of two main components: Boundary Distillation and Context Distillation. Below we present the detailed pseudo-code.

---

**Algorithm 1** Boundary and Context Knowledge Distillation (BCKD)

---

**Require:** Teacher model $T$, Student model $S$, Input image $X$

**Require:** Temperature $\tau$, Loss weights $\alpha$, $\beta$

**Require:** Maximum training epochs $t_{max}$

    **Feature extraction:**

1: $T_F \leftarrow \{T_i(X)\}_{i=1}^4$                                             ▷ Get teacher features

2: $S_F \leftarrow \{S_i(X)\}_{i=1}^4$                                            ▷ Get student features

3: $T_L, S_L \leftarrow$ logits from $T$ and $S$

    **Feature processing:**

4: $T_c \leftarrow \text{Conv}_{3\times3}(\text{concat}(T_1, \ldots, T_5))$

5: $S_c \leftarrow \text{Conv}_{3\times3}(\text{concat}(S_1, \ldots, S_5))$

    **Boundary Distillation:**

6: **for** each pair $(i,j)$ of pixels **do**

7:     $T_B^{i,j} \leftarrow 1 - \max_{p,q \in \Pi_{i,j}} B(\text{Conv}_{1\times1}(T_c^p), \text{Conv}_{1\times1}(T_c^q))$

8:     $S_B^{i,j} \leftarrow 1 - \max_{p,q \in \Pi_{i,j}} B(\text{Conv}_{1\times1}(S_c^p), \text{Conv}_{1\times1}(S_c^q))$

9: **end for**

10: $\mathcal{L}_{BD} \leftarrow -\tau^2 \sum_i \rho(T_B^i)^{1/\tau} \log(\rho(S_B^i)^{1/\tau})$

    **Context Distillation:**

11: $T_R \leftarrow \sigma\left(\frac{O(T_c)^T O(T_c)}{\sqrt{d}}\right)/\tau$

12: $S_R \leftarrow \sigma\left(\frac{O(S_c)^T O(S_c)}{\sqrt{d}}\right)/\tau$

13: $\mathcal{L}_{CD} \leftarrow -\tau^2 \sum_k \sigma(T_R^k)^{1/\tau} \log(\sigma(S_R^k)^{1/\tau})$

    **Loss computation:**

14: $r(t) \leftarrow 1 - (t-1)/t_{max}$

15: $\mathcal{L} \leftarrow \mathcal{L}_{SS} + r(t)(\alpha \mathcal{L}_{BD} + \beta \mathcal{L}_{CD})$

    **Model update:**

16: Update $S$ parameters via $\nabla \mathcal{L}$

**Ensure:** Trained student model $S$

---

