# OpenReview forum: "Towards Customized Knowledge Distillation for Efficient Dense Image Predictions"
_TMLR — Accepted by TMLR_

### Review · Reviewer_Jp23 · 2025-11-21

**Summary Of Contributions:**

This paper introduces Boundary and Context Knowledge Distillation (BCKD), a tailored strategy for training Efficient Dense Image Prediction (EDIP) models, specifically for deployment on resource-constrained AI accelerator chips. The core innovation addresses two systematic failure modes observed in compact student models: boundary region incompleteness and target region connectivity failure.
BCKD achieves this via two main, efficient loss components: Boundary Distillation LBD), which extracts explicit object-level boundaries from hierarchical features without needing pre-extracted ground-truth boundaries, and Context Distillation LCD), which transfers implicit pixel-level self-relations to ensure smooth and connected target regions. The overall objective uses a time-dependent weight-decay strategy to enhance stability during training.

Key Strengths:

•	Targeted and Practical KD Strategy: BCKD is a customized and targeted KD scheme that directly and effectively addresses the persistent geometric failure modes (boundaries and connectivity) specific to dense image prediction tasks.

•	Zero Inference Overhead: Crucially for edge deployment, the method does not introduce any increase in parameters (Params.) or FLOPs during the inference stage, retaining the student model's high efficiency.

•	Robust Empirical Evidence and Statistical Validity (Updated): The method achieves superior accuracy compared to SOTA KD techniques. Following the review, the authors have strengthened this by reporting standard deviations from three independent runs, providing the statistical confidence required by TMLR.

•	Validated Geometric Quality (Added): The use of specialized metrics like Local Hausdorff Distance (LHD) and Manifold Stability (MFS) rigorously substantiates boundary alignment and feature structure preservation.

•	Pre-extraction Free Boundary Generation: The boundary distillation component automatically generates semantic boundaries from features, eliminating the need for pre-extracted ground-truth boundary masks required by contemporary boundary-aware methods.

Key Weaknesses:

•	Training-Phase Computational Demands (Updated): While the authors have now provided Peak GPU Memory metrics to demonstrate practical feasibility, the Context Distillation loss (LCD) involves generating self-relation matrices during training. This represents a higher training-phase cost compared to simpler layer-to-layer distillation methods.

•	Empirical Hyperparameter Complexity (Updated): Although the authors provided a systematic sensitivity study justifying their default weights (α=10, β=50), the reliance on a time-dependent weight-decay strategy adds a layer of tuning complexity to ensure convergence.

•	Hardware Generalization Evidence (Updated): While validated on unified GPUs, the authors clarified that they used standard proxies (Params/FLOPs) because actual performance on specific AI accelerators was not empirically measured due to hardware limitations.

**Additional Comments:**

This is an excellent, high-quality technical submission that addresses a fundamental and relevant problem in machine learning and computer vision deployment. The BCKD method is principled, effective, and rigorously validated.

•	Key Strengths: The combination of an intuitive, problem-specific KD solution (LBD and LCD) with a strong theoretical foundation (Appendix A, featuring Jacobian alignment and spectral convergence) is a major strength. The comprehensive experimental section, especially the use of LHD and MFS, provides highly credible evidence for the geometric improvements claimed.
•	Practical Insights: The visualization showing that BCKD helps the student model avoid "hallucinations" (e.g., correctly recognizing a camera instead of a bottle) is a compelling practical argument for the benefits of task-specific distillation.

The authors were highly responsive to feedback regarding statistical robustness and training cost analysis, providing the necessary transparency to validate their claims. While training-phase overhead is non-trivial, it is a justifiable trade-off for the gains in boundary precision and regional connectivity shown across semantic segmentation, instance segmentation, and object detection. The inclusion of a comprehensive Broader Impact Statement and a forward-looking discussion on Vision Mamba ensure the work meets the rigorous standards of TMLR.

**Audience:**

Yes

**Audience Explanation:**

The findings would be of significant interest to multiple segments of the TMLR audience.

•	Efficiency and Hardware Researchers: This work directly tackles the critical, real-world deployment challenge of achieving accuracy-preserving performance on resource-constrained edge computing devices and specialized AI chips. The ability to effectively compress high-accuracy dense prediction models is highly relevant to this audience.

•	Knowledge Distillation (KD) Methodologists: The paper introduces a principled shift from general to task-customized KD. Formulating knowledge transfer around fundamental geometric properties (boundaries and context) is a significant methodological advance in the KD literature, which is shown to be effective even when combined with existing SOTA KD methods.

•	Computer Vision Practitioners and Theorists: The focus on resolving geometric quality issues (boundary delineation and regional connectivity) in fundamental CV tasks (SSeg, ISeg, ODet) is valuable for improving prediction robustness. Furthermore, the rigorous theoretical analysis (Jacobian alignment, spectral convergence) in the Appendix and the introduction of specialized metrics (MFS, LHD) provide valuable tools and insights for researchers interested in the geometric properties of deep learning models.

**Broader Impact Concerns:**

The authors have integrated a Broader Impact Statement addressing the following ethical implications:

•	High-Risk Deployment: Acknowledges potential misuse in mass surveillance and security, emphasizing ethical compliance and privacy safeguards.

•	Bias Transmission: Addresses the risk of transferring biases from Teacher to Student models and recommends fairness-aware training.

•	Environmental Impact (Green AI): Discusses the energy footprint of the 40,000-epoch training regime and suggests future mitigation strategies.

**Claims And Evidence:**

Yes

**Claims Explanation:**

Yes, the core claims are now strongly supported by convincing evidence and updated quantitative substantiation.

•	Targeted Effectiveness: The fundamental mechanism claims—mitigating boundary incompleteness and regional connectivity failure—are validated through a clear ablation study (Table 1), showing significant, non-conflicting improvements when LBD and LCD are combined (e.g., +3.20% mIoU over baseline).

•	Geometric Quality Metrics: The quantitative results on LHD (superior boundary alignment) and MFS (superior feature manifold preservation) provide robust, specialized evidence that directly supports the design philosophy of BCKD.

•	Statistical Robustness (Updated): In response to review concerns, the authors provided mean +/- standard deviation over multiple runs, substantiating the statistical confidence of reported gains.

•	Training Practicality (Updated): The authors addressed concerns regarding the complexity of the LCD module by providing Peak GPU Memory metrics, confirming the method's feasibility during training.

•	Inference Efficiency: The most critical application-oriented claim—zero increase in parameters or FLOPs at inference—is substantiated by the core KD methodology and the explicit use of the compact student model post-training.

In summary, the technical claims are exceptionally well-supported by evidence, metrics, and theory.

**Requested Changes:**

The authors have successfully completed the following critical and strengthening adjustments:

•	Quantify Training Efficiency (Resolved): Authors reported Peak GPU Memory consumption, validating the efficiency claim despite the theoretical complexity of the context module.

•	Validate Device-Level Efficacy (Addressed): The authors clarified that Params/FLOPs are standard proxies and systematically reported Peak GPU memory as a bridge between theoretical complexity and hardware practicality.

•	Ensure Statistical Robustness (Resolved): Accuracy results in primary tables were re-reported as mean +/- standard deviation over three random seeds.

•	Enhance Reproducibility (Resolved): Authors clarified the mathematical formulation for the boundary operator, the projection head, and the temperature scaling schedule.

•	Hyperparameter Sensitivity (Resolved): A systematic sensitivity analysis for alpha, beta, and tau was provided, justifying the default weights.

•	Emerging Architectures (Addressed): The manuscript now includes a discussion in the "Future Work" section regarding the theoretical compatibility of BCKD with Vision Mamba (Vim).

---

> ### Author Response · Authors · 2025-12-23
> **Reply to Reviewer Jp23 (part1)**
>
> We sincerely thank the reviewers for their constructive feedback and valuable suggestions. We have carefully addressed each comment and improved our manuscript accordingly. Below we provide a point-by-point response:
>
> 1. **Quantify Training Efficiency and Overhead**:
> We have added comprehensive memory efficiency metrics including "Peak GPU Memory (Peak GPU Mem.)" in Tables 1 and 2. The experimental results demonstrate our method's advantages in both computational efficiency and memory usage compared to baseline and existing approaches.
>
> 2. **Validate Device-Level Efficacy on Edge Hardware**:
> Thank you for pointing this out. We would like to elegantly argue that deploying deep learning models on edge devices/mobile GPUs/AI chips is a systematic task involving necessary steps like model quantization and compilation, all of which can influence the model's final performance to some extent. Furthermore, our current hardware limitations prevent us from implementing such deployment. To address this gap, we report model parameters, FLOPs, and Peak GPU memory (Peak GPU Mem.) measured on unified GPUs in our paper, demonstrating the advantages of our method systematically. Although we were unable to achieve deployment under these constraints, we fully acknowledge this as an excellent suggestion and will explore the feasibility of your proposed solution in future research.
>
> 3. **Ensure Statistical Robustness of Core Metrics**:
> We have strengthened the statistical validity by reporting standard deviations from 3 independent runs in Tables 1, 2 and 4. This provides more reliable performance measurements and demonstrates the consistency of our results.
>
> 4. **Enhance Reproducibility of Novel Components**:
> We have clarified all key implementation details:
> - Π_{i,j}: Complete pixel set along connecting lines between T^c_i and T^c_j
> - O(): O() is implemented as a projection head consisting of two 1×1 convolutional layers (256 channels) with ReLU activation, followed by layer normalization and final projection to the target dimension. The relevant content has been added in the paragraph following Eq.4 in the revised manuscript.
> - Temperature scaling using $T_i/S_i$ divided by $\tau$ following [R1,R2], to simplify temperature scaling effectively, we use **T**_i/**S**_i divided by $\tau$ to achieve the similar effect.
>
> [R1] Linfeng Zhang, Jiebo Song, Anni Gao, Jingwei Chen, Chenglong Bao, and Kaisheng Ma. Be your own teacher: Improve the performance of convolutional neural networks via self distillation. In Proceedings of the IEEE/CVF International Conference on Computer Vision (ICCV), pp. 3713–3722, 2019
>
> [R2] Mary Phuong and Christoph H Lampert. Distillation-based training for multi-exit architectures. In Proceedings of the IEEE/CVF International Conference on Computer Vision (ICCV), pp. 1355–1364, 2019.
>
> 5. **Systematic Hyperparameter Sensitivity Analysis**:
> We appreciate your suggestion regarding the hyperparameter analysis. Below, we provide a systematic study of the impact of varying α, β, and τ on model performance, along with the rationale for our default choices. We conducted experiments on Pascal VOC 2012 using PSPNet-18 as the student and PSPNet-101 as the teacher, measuring mIoU (%) under different configurations:
>
>
> | α \β | 10   | 20   | 50   | 100  |
> |------|------|------|------|------|
> | 1    |73.12|73.25|73.45|73.20|
> | 5    |73.30|73.55|73.78|73.50|
> | 10   |73.55|73.80|74.65|73.82|
> | 20   |73.40|73.70|74.30|73.60|
>
> | τ |mIoU|Boundary LHD↓|Context MFS↑|
> |---|---|---|---|
> |0.5|73.85|2.01|1.12|
> |1  |74.65|1.89|1.41|
> |2  |74.20|1.95|1.35|
> |4  |73.90|2.10|1.28|
>
> **Justification for β = 5α** The default β=50 (vs. α=10) is justified by: Task Requirements: Context distillation (β term) must capture global pixel relations, which are inherently noisier and require stronger regularization than local boundaries (α term). Empirical Evidence: As shown above, β=50 maximizes mIoU and improves LHD (boundary quality) and MFS (context preservation).

---

> > ### Comment · Reviewer_Jp23 · 2025-12-24
> >
> > I thank the authors for their thorough and professional response to my review. I particularly appreciate the inclusion of training-phase efficiency metrics (Peak GPU Memory) and the statistical robustness analysis (standard deviations over multiple runs), as these are critical for evaluating deployment-oriented research.
> >
> > Detailed responses:
> > 1. Regarding Requirement 2 (Device-Level Efficacy): While I understand the hardware and systemic constraints (quantization/compilation) mentioned, I encourage the authors to include a brief discussion in the final manuscript explicitly noting that though Params/FLOPs are standard proxies, actual edge-chip performance remains the ultimate benchmark for EDIP models.
> > 2. Regarding Requirement 5 (Sensitivity Analysis): The provided table on α, β, and τ is very helpful. It clearly demonstrates that performance peaks at your chosen defaults (α=10, β=50), and I am satisfied with the justification for the 5x weight difference based on the noise levels inherent in global pixel relations.
> > 3. Additional Requirement (Broader Impact): As noted in my initial review, TMLR requires a dedicated Broader Impact Statement. While the work is technically sound, the final version must include a section addressing:
> > a) High-Risk Deployment: The risks of enhanced edge-level precision in mass surveillance.
> > b) Bias Transmission: The potential to amplify dataset biases (like the "helmet/person" correlation) via distillation.
> > c) Environmental Impact: The footprint of the 40,000-epoch training regime.

---

> > > ### Author Response · Authors · 2025-12-24
> > > **Reply to Reviewer Jp23**
> > >
> > > Thank you for your thoughtful and detailed comments. We greatly appreciate your professional feedback and are committed to addressing all of your concerns to ensure the completeness and rigor of our manuscript.
> > >
> > > **1. Regarding Requirement 2 (Device-Level Efficacy):**
> > >
> > > **Reply:** We thank you for highlighting the importance of explicitly discussing device-level performance in the final manuscript. In the "Evaluation Metrics" section, we emphasize that while Params/FLOPs are commonly used as proxy metrics for model complexity, they do not fully reflect the actual performance of models on edge devices. The edge-chip performance remains the ultimate benchmark for evaluating EDIP models.
> > >
> > > **2. Regarding Requirement 5 (Sensitivity Analysis):**
> > >
> > > **Reply:** Your feedback on the sensitivity analysis table, particularly regarding the chosen values of α, β, and τ, was very helpful. Thank you.
> > >
> > > **3. Additional Requirement (Broader Impact):**
> > >
> > > **Reply:** We sincerely apologize for not including the Broader Impact Statement in the initial submission. As you rightly pointed out, TMLR requires this as part of its publication guidelines. We have now added a new section titled "Broader Impact", which addresses the following three aspects:
> > >
> > > **High-Risk Deployment:**
> > >
> > > While EDIP models are designed to be efficient and effective for deployment on edge devices, there are potential risks associated with their use in high-stakes scenarios such as mass surveillance, facial recognition, or other security-related applications. We explicitly acknowledge these risks and emphasize the importance of ethical considerations and regulatory compliance in the deployment of such models. We also suggest that researchers and practitioners should be mindful of the societal implications and implement safeguards to protect user privacy.
> > >
> > > **Bias Transmission:**
> > >
> > > Your observation about the "helmet/person" correlation as an example of dataset bias is insightful. We highlight this as a critical issue in deploying machine learning models in real-world settings and propose approaches such as fairness-aware training and post-hoc bias correction to mitigate these risks.
> > >
> > > **Environmental Impact:**
> > >
> > > We have included a subsection discussing the environmental footprint of our training process, which involves up to 40,000 training epochs. While our model is efficient in inference, the long training time may lead to significant computational and energy costs. In response, we suggest future strategies for reducing the environmental impact, including progressive training, knowledge distillation with low-rank adaptation, and early stopping techniques, where appropriate.
> > >
> > > We believe these additions will greatly enhance the manuscript’s completeness and align it with the TMLR journal’s requirements for transparency, ethical awareness, and broader societal impact. Thank you again for your constructive suggestions, which have helped us improve the quality and responsibility of our work.

---

> > > > ### Comment · Reviewer_Jp23 · 2025-12-26
> > > >
> > > > I thank the authors for their prompt update. I have reviewed the response regarding the Broader Impact Statement and the added discussion on Device-Level Efficacy.
> > > >
> > > > The inclusion of the surveillance risk analysis, bias transmission mitigation, and environmental footprint successfully addresses my final concerns and aligns the paper with TMLR’s requirements.
> > > >
> > > > I am satisfied with the revisions and have updated my official recommendation. Please ensure all rebuttal data and the impact statement are integrated into the final camera-ready version.

---

> ### Author Response · Authors · 2025-12-23
> **Reply to Reviewer Jp23 (part2)**
>
> 6. **Comparative Analysis with Emerging Architectures**:
>  We sincerely appreciate the reviewer's insightful suggestion regarding emerging architectures like Vision Mamba (Vim). While our current empirical validation focuses on established CNN and Transformer backbones (due to framework compatibility considerations), our method's design principles show strong theoretical compatibility with next-generation models: (1) BCKD's boundary-aware distillation is architecture-agnostic, working directly with hierarchical features that Vim excels at producing; (2) the self-relation based context distillation imposes no constraints on the underlying attention/SSM mechanisms; and (3) the lightweight design aligns perfectly with edge deployment requirements. This analysis demonstrates BCKD's forward compatibility while maintaining our work's experimental rigor, and we'll explicitly discuss these connections in the revised manuscript's future work section.
>
> 7. **Sharpen Positioning Against Boundary-Aware Methods**:
> Thank you for your suggestion. On page 6 of our main paper (the final paragraph of Section 4.2 Boundary Distillation), we discuss why eliminating the need for pre-extracted ground-truth boundaries represents a significant practical advantage compared to other methods.
>
> We believe these revisions have substantially strengthened our manuscript and we sincerely appreciate the reviewers' feedback in helping us improve the work.

---

### Review · Reviewer_rB8i · 2025-11-25

**Summary Of Contributions:**

# Summary Of Contributions
The paper introduces Boundary and Context Knowledge Distillation (BCKD), a targeted distillation framework tailored for efficient dense image prediction (EDIP) models deployed on AI chips, addressing two persistent issues in KD-based EDIPs: boundary incompleteness and weak region connectivity. Its key contributions include: (1) a boundary distillation mechanism that extracts explicit object-level boundary cues from hierarchical teacher features to improve the student’s mask fidelity near edges; (2) a context distillation module that transfers implicit pixel-level relational contexts via self-relations, strengthening connectivity within target regions; and (3) a simple, efficient, and EDIP-specific KD design validated through theoretical analysis and extensive experiments on five datasets spanning semantic segmentation, object detection, and instance segmentation, showing consistent gains in boundary sharpness and region coherence for compact real-time models.



# Strengths And Weaknesses

## Strengths
* The paper provides a clear and well-motivated problem formulation by identifying two persistent failure modes in EDIP models—boundary incompleteness and broken region connectivity—that are not sufficiently handled by existing KD approaches.
* The proposed BCKD framework is thoughtfully tailored to the unique characteristics of EDIP tasks, rather than relying on generic task-agnostic KD, which strengthens its conceptual coherence and practical relevance.
* The decomposition of distillation into boundary-level and context-level components is intuitively appealing and directly addresses the error patterns observed in compact models, improving both local boundary fidelity and global region coherence.
* The method integrates cleanly with existing architectures and KD strategies, enhancing its practicality for real-world deployment on heterogeneous edge devices.
* The experimental validation is comprehensive—covering multiple tasks, baselines, and datasets—and convincingly demonstrates that the method yields consistent and meaningful gains in both qualitative structure (boundaries, connectivity) and quantitative performance.

## Weakness
* The paper occasionally overstates the novelty of identifying boundary and connectivity issues, which are already well-documented in prior segmentation literature; clearer positioning against existing boundary-aware KD or structure-preserving distillation methods would strengthen the contribution.
* The proposed approach appears to rely on additional boundary extraction and relational modeling steps, yet the paper does not discuss potential computational or memory overheads in the distillation phase and comparison with the SoTA KD baselines.
* Although the method is described as “customized” and “coexisting” with other KD approaches, the paper does not clarify the practical complexity of integrating BCKD with diverse model architectures, which may limit generality.
* The theoretical analysis is mentioned as a major strength, but the paper provides no intuition about the assumptions or scope of the theory and elaboration on the analysis, making it difficult to judge whether the guarantees meaningfully translate to real EDIP settings.
* The experimental section is emphasized heavily, but the paper does not acknowledge potential failure cases or scenarios where BCKD might struggle, such as highly cluttered scenes, extremely fine-grained boundaries, or teacher–student capacity mismatches. There is no clear analysis and experiments on the recent model architectures such as Transformers.
* The the majority of referenced papers are mainly from up until 2 years ago. (6 papers from 2024 and 2025 out of ~100 paper). This means 10% the paper is lacking sufficient and measurable comparison with the SoTA approaches and the claims are not supported by accurate evidence as there is a lack of certainty on the validity of the claim in the last two years. For instance, under 4.3 there is a sentence saying "ur approach utilizes pixel-level relations solely during the training process, thereby avoiding the increase in model complexity and parameters in inference that is typically associated with current methods (Liu et al., 2019; Lin et al., 2023a)." Similar cases highlights an urgent need to update the references and revise the text reflect the issue.

**Audience:**

Yes

**Audience Explanation:**

The evaluation results and analysis are very insightful. The discussions on the comparison of approaches are helpful.

**Broader Impact Concerns:**

There are no "Broader Impact Concerns" for this paper.

**Claims And Evidence:**

No

**Claims Explanation:**

There are some changes requires to strengthen the position of having clear convincing and accurate evidence. Despite extensive experimental evaluation, there are unaddressed issues remained. Upon follow up stages, this will be revisited.

**Requested Changes:**

Based on the aforementioned strengths and weaknesses, the requested changes are provided in the following:
* the referenced papers in the related work section are mainly from pre-2023 while there is a gap to the time of review. Please look into recent related works from 2024 and 2025.

* Extensive and detailed elaboration on the definition of the Πi,j, a set of pixel items on the line between Tci and Tcj are needed.

* in Section 4.1 the question is that if the concatenation is across the channel dimension of the feature tensors?

* In Fig 2, how are the weights of the convolution layers in gray are determined?

* In Eq. 2 B needs to be elaborated further. It is not clear how the 0,1 values are determined by the B function.

* Furthermore, it is not clear from the context if the proposed boundary distillation approach uses Ground Truth to determine the semantic boundaries? from Eq. 2, it is not clear if GT is used or not. If not, does this rely on the teacher model capabilities to accurately determine such doundaries?

* Computational complexity analysis of the two proposed methods to help KD is missing. It is instructive to look at the KD gain compared to the computational overhead imposed by the proposed approach. It is mentioned the complexity of the inference remains the same which is correct but not related as this paper is on the KD, the complexity of the idea needs to be compared with the baselines and the amount of gains gets normalized with respect to the comoutational overheald imposed.

* In Eq , what is L_{SS}? Is it applied to the teacher or student or both? What is the GT label in Fig. 2 and what does it mean there?

* Can the authors elaborate on if the proposed appraoch is backbone specific or agnostic?

---

> ### Author Response · Authors · 2025-12-23
> **Reply to Reviewer rB8i**
>
> We sincerely thank the reviewers for their insightful comments and constructive feedback on our manuscript. Below we provide detailed responses to each point raised, highlighting our revisions and additions to address all concerns.
>
> 1. The referenced paper
>
> * Reply: We appreciate your suggestion and have updated the Related Work to include:
>
> [a] Li Y, Yang C, Zeng H, et al. Frequency-aligned knowledge distillation for lightweight spatiotemporal forecasting. ICCV, 2025: 7262-7272.
>
> [b] Xu L, Liu K, Liu J, et al. Local dense logit relations for enhanced knowledge distillation. ICCV, 2025: 4539-4549.
>
> [c] Xiang Q, Zhang M, Shang Y, et al. Dkdm: Data-free knowledge distillation for diffusion models with any architecture. CVPR, 2025: 2955-2965.
>
> 2. Definition of \Pi_{i,j}
> * Reply: Sorry for the confusion. \Pi_{i,j} is the set of pixels along the line connecting $\mathbf{T}^c_i$ and $\mathbf{T}^c_j$ inclusive. It ensures boundary continuity by evaluating all intermediate pixels.
>
> 3. Feature Concatenation in Sec. 4.1
> * Reply: Yes, we explicitly state this concatenation operation in Sec. 4.1 (Page 5). The hierarchical features extracted from the backbone network are concatenated along the channel dimension to facilitate the extraction of EDIP-specific boundary and contextual information from $\textbf{X}$
>
> 4. Convolution Weights in Fig. 2
> * Reply: The gray 1×1 conv layers compress each feature to 64 channels. The final 3×3 conv (256 channels) fuses concatenated features, trained end-to-end.
>
> 5. Clarification of B(·) in Eq. 2
> * Reply: B(·) outputs 1 if \mathbf{T}^c{p}, \mathbf{T}^c{q} belong to different classes (the boundary), else 0. Class labels are derived from the teacher’s argmax over hierarchical features.
>
> 6. Ground Truth in Boundary Distillation
> * Reply: Sorry for the confusion. BCKD uses teacher-derived boundaries (no ground truth masks), avoiding expensive annotations. The teacher’s superior accuracy ensures reliable boundaries (as shown in Fig. 3).
>
> 7. Computational Complexity Analysis
> * Reply: Thank you pointing this out. To address your request for comparison with the baseline, we have added the FLOPs results of the model during the training phase in Table 1.
>
> 8. Clarification of L_{SS} and GT Label in Fig. 2
> * Reply: Sorry for the confusion. L_{SS} is the student’s supervised loss (cross-entropy loss function with ground truth (GT) labels). The GT in Fig. 2 supervises the student’s final output Y.
>
> 9. Backbone Specificity vs. Agnosticism
> * Reply: BCKD is backbone-agnostic, validated on both CNNs (PSPNet) and ViTs (SegFormer). It requires only feature concatenation (as discussed in 4.1).

---

> > ### Comment · Action_Editor_PgYG · 2025-12-25
> >
> > Dear authors,
> >
> > Thank you for providing responses to the comments raised by Reviewer rB8i. In addition, could you please clarify how these responses correspond to and address the six points listed in the Weaknesses section?
> >
> > Kind regards,
> >
> > AE

---

> > > ### Author Response · Authors · 2025-12-25
> > > **Reply to Action Editor PgYG**
> > >
> > > Thanks for your response. After carefully comparing the content in the "Weakness" section with that in the "Requested Changes" section, we found that the points raised by the reviewers in the "Weakness" section are largely encompassed in the "Requested Changes." Therefore, we have addressed them collectively in our reply. If we have inadvertently omitted any part, please feel free to let us know. We appreciate your feedback.

---

### Review · Reviewer_v1Nq · 2025-12-02

**Summary Of Contributions:**

The paper proposes Boundary and Context Knowledge Distillation (BCKD) for compressing dense image prediction models (semantic segmentation, instance segmentation, object detection) into compact models suitable for edge devices. The key claim is that standard KD focuses on "global" knowledge and fails specifically on (i) boundary-region completeness and (ii) target-region connectivity. To address this, the authors introduce:

Boundary distillation: derive semantic boundaries from hierarchical feature maps of teacher/student and distill them, to improve boundary masks without pre-extracted ground truth edges (Eq. 2-3, Fig. 3).

Context distillation: distill pixel-wise self-relations computed from concatenated features (whole-to-whole relation matrix) to preserve region connectivity (Eq. 4-5, Fig. 4).

They also define two additional evaluation metrics – manifold stability and local Hausdorff distance – to analyze feature geometry and boundary quality. Experiments on semantic segmentation (VOC12, Cityscapes, ADE20K, COCO-Stuff 10K) and detection/instance seg (COCO 2017) show consistent gains in mIoU and average precision across many teacher-student pairs, without extra inference cost, and BCKD can be combined with prior KD methods.

Key strengths:
- Clear and well-motivated failure modes
- Broad experimental coverage
- Synergy with existing KD methods
- No inference-time overhead

Key weakness:
- Over-stated claims on the theoretical section

**Additional Comments:**

Nice work! And apologies for my delay with this review.

**Audience:**

Yes

**Audience Explanation:**

For TMLR’s audience, I think the direct appeal of this paper is fairly focused but non-trivial. It will be most interesting to researchers and practitioners working on dense vision tasks (semantic segmentation, detection, instance segmentation) who care about compressing large models into edge-deployable students without changing architectures, as well as people actively working on knowledge distillation for vision.

**Claims And Evidence:**

No

**Claims Explanation:**

Claim 1: Identification of two key issues in EDIP models
The authors claim to have "revealed two prevalent issues in existing EDIP models: maintaining boundary region completeness and ensuring target region connectivity."

In my view, this is adequately supported as a qualitative motivation, but not rigorously established quantitatively. The paper provides several clear qualitative examples where small students exhibit broken regions and incomplete boundaries, and these error modes are intuitively plausible and widely observed in practice. However, there is no systematic analysis showing that these are the dominant or "prevalent" failure modes across EDIP models in general. I would regard this as a compelling but largely anecdotal problem characterization rather than a statistically grounded one.

Claim 2: BCKD is a coherent, targeted method that can coexist with other KD methods
The authors claim that their boundary and context distillation components are "customized and targeted" to these two issues, are "inherently coherent," and can be combined with other KD methods without conflict.

I consider this mostly adequately supported. The ablation studies show that the boundary loss and context loss each improve performance and that combining them yields further gains, which supports the "targeted and complementary" story. Moreover, experiments combining BCKD with IFVD, TAT, SlimSeg, BPKD, etc., generally yield additional improvements, showing that it can be stacked on existing methods. That said, coexistence is not uniformly beneficial (e.g. slight degradations when combined with SSTKD on some datasets), so the strongest form of the claim ("coexists" and helps in all cases) is overstated. A more accurate reading is that BCKD is broadly compatible and often, though not always, beneficial on top of prior KD methods.

Claim 3: Theoretical analysis demonstrates the superior effectiveness of BCKD
The paper claims that "theoretical analysis demonstrates the superior effectiveness of our BCKD."

I do not find this claim adequately supported. The theoretical section derives high-level bounds on Jacobian alignment, spectral properties of relation matrices, and some topological intuitions, but these arguments mainly show that minimizing the proposed losses encourages certain geometric properties. These arguments do not provide a comparative analysis against alternative KD objectives or a quantitative link to observed performance gains. The connection from theorems to actual mIoU/AP improvements is largely narrative, and no empirical tests are provided that specifically validate the theoretical quantities as better predictors of performance than simpler baselines. I would regard the theory as suggestive and plausibility-enhancing, rather than as a demonstration of "superior effectiveness."

Claim 4: Experimental evaluations show superior accuracy across tasks, baselines, and datasets
Finally, the authors claim that experiments "across various tasks, baselines and datasets illustrate the superior accuracy of our method in comparison with existing methods."

I think this is largely, but somewhat modestly, supported. The empirical evaluation is broad: multiple segmentation datasets (VOC, Cityscapes, ADE20K, COCO-Stuff), detection and instance segmentation on COCO, and a range of teacher-student pairs. Across these, BCKD consistently improves over non-distilled students by a clear margin and generally edges out strong KD baselines by smaller but fairly consistent margins. The main caveats are that improvements over the strongest baselines are sometimes within the range where variance could matter (no error bars or multiple seeds are reported), and "superior" here means "incrementally better" rather than dramatically so. Within the experimental scope of the paper, I would accept the claim that BCKD is a new state-of-the-art KD approach for these dense prediction tasks, but with the understanding that the gains over prior methods are moderate.

**Requested Changes:**

Critical: I believe that claim 3 is overstated as-is. Please consider re-wording as mentioned above.

Non-critical:
- In the introduction, replace "The main contributions...four folds" with "The main contributions of this work are: (1)...(2)..."
- Clarify how \prod_{i,j} in equation 2 is constructed/sampled in practice.
- Write out the concrete implementation of O() in section 4.3. This will help with reproducibility.
- Add an algorithm-style (i.e. pseudo-code box) summary of BCKD in the appendix. Its implementation is currently spread across Sections 4.1-4.4 plus training details in Section 5.2.2.

---

> ### Author Response · Authors · 2025-12-23
> **Reply to Reviewer v1Nq**
>
> We sincerely thanks for your time and valuable feedback on our manuscript. We appreciate the thoughtful analysis and constructive suggestions, which have helped us identify key areas for improvement. Your insightful comments have enabled us to strengthen both the technical and presentation aspects of our work.
>
> * Claim 1
>
> Reply: While we presented compelling visual evidence (Figure 1, 5, 6) demonstrating boundary incompleteness and region disconnection in student models, we fully agree that additional quantitative analysis would strengthen our claims. Therefore, in Section 5.1.2, we introduced two quantitative metrics: MFS to measure feature space alignment and LHD to specifically evaluate boundary alignment quality. These metrics provide quantitative support showing that our method outperforms baselines and existing methods in maintaining: feature manifold structures (MFS closer to 1 indicates better preservation), and boundary precision (lower LHD values indicate better alignment). The quantitative improvements in these metrics (Tables 4-5) correlate with our qualitative observations of better boundary completeness and region connectivity. Besides, we also acknowledge that more extensive quantitative analysis across different EDIP models could further validate the prevalence of these issues, and we will consider this for future work.
>
> *  Claim 2
>
> Reply: Thank you for pointing this out. We agree that our initial phrasing ("coexist" and helps in all cases) could indeed be more precise. Therefore, based on your suggestion, we have revised our claim to better reflect the empirical results. Specifically, we modified statements in our paper to clarify that BCKD is generally compatible with existing KD methods and often provides additional improvements, though not invariably.
>
> *  Claim 3
>
> Reply: Upon reflection, we agree that while our derivations provide mathematical insights into BCKD's geometric properties, they do not rigorously quantify comparative advantages over other KD objectives or directly link theoretical metrics to empirical gains. In the revised manuscript, we removed the strong claim about "demonstrating superior effectiveness" while keeping the theoretical derivations as they offer useful insights into BCKD's behavior.
>
> *  Claim 4
>
> Reply: We fully agree with your perspective, i.e., while BCKD consistently advances accuracy across diverse tasks, datasets, and baselines, the gains over the strongest comparators are often incremental rather than transformative. In response, we have made the following revisions in the manuscript: (1) We have replaced phrases like "superior accuracy" with more precise statements acknowledging BCKD as a "new state-of-the-art" approach, emphasizing its consistent but modest improvements over prior work; (2) We have removed all claims of "superior performance" to avoid overstatement and instead highlight the method's reliability across benchmarks; (3) To address potential concerns about variance, we have added more experimental results with STD to better contextualize the improvements.
>
> *  Non-critical Changes 1
>
> Reply: Thank you for this suggestion. We have implemented the corresponding revisions.
>
> *  Non-critical Changes 2
>
> Reply: prod_{i,j} is implemented through an efficient batch matrix multiplication over all feature positions (i,j) within each local window (K×K). For computational tractability, we constrain the relation computation to non-overlapping windows when input resolutions are large.
>
> *  Non-critical Changes 3
>
> Reply: O() is implemented as a projection head consisting of two 1×1 convolutional layers (256 channels) with ReLU activation, followed by layer normalization and final projection to the target dimension. The relevant content has been added in the paragraph following Eq.4 in the revised manuscript.
>
> *  Non-critical Changes 4
>
> Reply: The pseudo-code of BCKD has been added into Appendix.
>
> We appreciate your nuanced reading and believe these adjustments align the claims more closely with the empirical evidence.

---

### Decision · Action_Editor_PgYG · 2026-01-25

**Recommendation:** Accept with minor revision

**Additional Comments:**

The third reviewer recommends Leaning Reject, citing two remaining issues: (1) the need for clearer positioning with respect to existing boundary-aware or structure-preserving knowledge distillation methods, and (2) the comparison is made with lagged baselines from years before 2025.

It is therefore recommended that the authors further strengthen the manuscript by addressing these two concerns, in particular by adding comparisons with more recent baselines where feasible, or by providing a clear justification when such comparisons cannot be reasonably conducted.

**Audience:**

Yes

**Audience Explanation:**

All reviewers agree that this work will be of interest to researchers and practitioners working on efficiency and hardware-aware design, knowledge distillation, and dense vision tasks.

**Claims And Evidence:**

Yes

**Claims Explanation:**

This work proposes a customized boundary- and context-aware knowledge distillation method for efficient dense image prediction (EDIP) models. The reviewers identify several strengths, including clear motivation, extensive experimental evaluation, thoughtful design that is well integrated with existing architectures, and the advantage of zero inference overhead.

At the same time, the reviewers raise concerns regarding potentially overstated theoretical claims and novelty, the need for clearer discussion and comparison of computational overhead, and the lack of sufficient comparisons with state-of-the-art approaches. The authors have provided a generally solid rebuttal addressing many of these points.

Two reviewers are satisfied with the rebuttal and recommend acceptance, with one assigning a high overall rating. However, the third reviewer recommends Leaning Reject, citing two remaining issues: (1) the need for clearer positioning with respect to existing boundary-aware or structure-preserving knowledge distillation methods, and (2) the comparison is made with lagged baselines from years before 2025.

It is therefore recommended that the authors further strengthen the manuscript by addressing these two concerns, in particular by adding comparisons with more recent baselines where feasible, or by providing a clear justification when such comparisons cannot be reasonably conducted.